# LoRA-X: Bridging Foundation Models with Training-Free Cross-Model Adaptation

**Farzad Farhadzadeh**[*]**, Debasmit Das, Shubhankar Borse, Fatih Porikli**
Qualcomm AI Research[†], San Diego, CA 92121, USA
{ffarhadz, debadas, sborse, fporikli}@qti.qualcomm.com

## Abstract

The rising popularity of large foundation models has led to a heightened demand for parameter-efficient fine-tuning methods, such as Low-Rank Adaptation (LoRA), which offer performance comparable to full model fine-tuning while requiring only a few additional parameters tailored to the specific base model. When such base models are deprecated and replaced, all associated LoRA modules must be retrained, requiring access to either the original training data or a substantial amount of synthetic data that mirrors the original distribution. However, the original data is often inaccessible due to privacy or licensing issues, and generating synthetic data may be impractical and insufficiently representative. These factors complicate the fine-tuning process considerably. To address this challenge, we introduce a new adapter, Cross-Model Low-Rank Adaptation (LoRA-X), which enables the training-free transfer of LoRA parameters across source and target models, eliminating the need for original or synthetic training data. Our approach imposes the adapter to operate within the subspace of the source base model. This constraint is necessary because our prior knowledge of the target model is limited to its weights, and the criteria for ensuring the adapter's transferability are restricted to the target base model's weights and subspace. To facilitate the transfer of LoRA parameters of the source model to a target model, we employ the adapter only in the layers of the target model that exhibit an acceptable level of subspace similarity. Our extensive experiments demonstrate the effectiveness of LoRA-X for text-to-image generation, including Stable Diffusion v1.5 and Stable Diffusion XL.

## 1 Introduction

Large foundation models (LFMs) have demonstrated outstanding performance across various domains, including natural language processing (OpenAI et al., 2023; Gemini Team et al., 2023; Anthropic, 2024; AI@Meta, 2024) and computer vision (Ho et al., 2020; Rombach et al., 2022). Due to their remarkable capabilities, fine-tuning LFMs for a wide array of downstream tasks has become common practice. In the full fine-tuning approach, each new model tailored to a specific task generally retains the same number of parameters as the original model. As models increase in size and customization requirements grow, the need to store such fully fine-tuned checkpoints also rises, leading to substantial storage and memory costs.

To address this challenge, Parameter-Efficient Fine-Tuning (PEFT) (Xu et al., 2023) methods, such as Low-Rank Adaptation (LoRA) (Hu et al., 2022), offer a promising solution. PEFT methods aim to reduce the number of parameters that need to be updated during fine-tuning, making the process more computationally efficient and less resource-intensive.

LoRA, in particular, addresses this by representing the weight changes in the model using two low-rank matrices, $A$ and $B$. Specifically, the weight update is expressed as $W_0 +$

---

[*]Corresponding author
[†]Qualcomm AI Research is an initiative of Qualcomm Technologies, Inc.

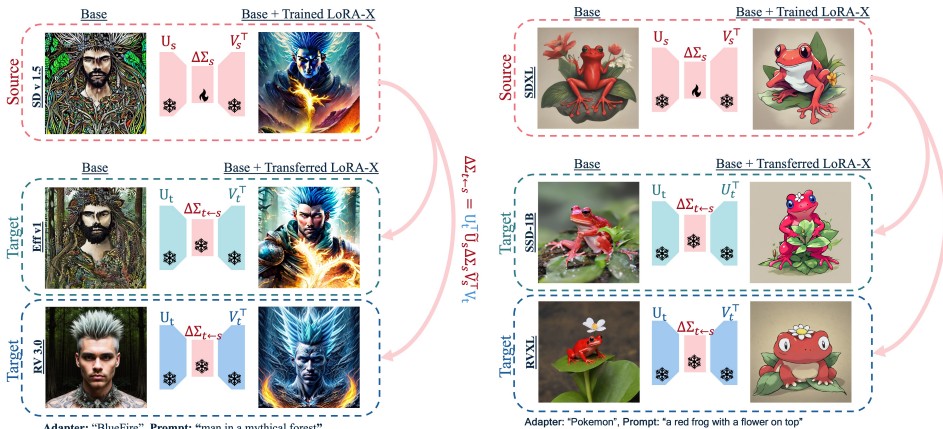

Figure 1: LoRA-X training-free transfer (a) from source SD-v1.5 to targets SD Eff-v1.0 and RV-v3.0, (b) from source SDXL to targets SSD-1B and RVXL-v3.0.

$\Delta \boldsymbol{W} = \boldsymbol{W}_0 + \boldsymbol{B}\boldsymbol{A}$, where $\boldsymbol{W}_0$ is the original weight matrix, and $\Delta \boldsymbol{W}$ is the change applied during fine-tuning. By decomposing the weight changes into these low-rank matrices, LoRA significantly reduces the number of parameters that need to be adjusted, thus making the fine-tuning process more efficient.

Nevertheless, a consequential limitation of the LoRA approach is its dependency on the base model. A LoRA adapter fine-tuned for a specific task is intrinsically tied to its base model and cannot function independently of it. Moreover, it cannot be directly transferred to another base model without additional training. This dependency poses a profound challenge when the base model is deprecated and replaced by a newer version. In such cases, applications utilizing LoRA adapters from the deprecated model must transition their adapters to the newer base model. This process can be cumbersome, resource-intensive, and, crucially, infeasible if the original fine-tuning data is no longer available.

This naturally leads us to an important question: *Can we design a LoRA adapter that can be transferred between different base models without requiring additional training or access to the original data?* A solution addressing this problem allows the longevity of LoRA adapters and eliminates the need for repeated fine-tunings as base models evolve.

In this work, we introduce LoRA-X, a compact adapter that can be transferred between different versions of a base model without the need for fine-tuning with the original data. The core idea behind LoRA-X is to maintain the adapter within the column-row subspace of the base model weights. This innovative strategy ensures that the adapter remains compatible with alternative versions of the base model, simplifying the transition process and enhancing the adaptability of fine-tuned models.

To the best of our knowledge, LoRA-X is the first adapter designed for transferability without additional training. Our main contributions are (See Figure 1):

- **Concept of LoRA-X**: We introduce LoRA-X, a versatile cross-model adapter that can be seamlessly transferred across various base models without the need for additional training. This innovation simplifies the process of adapting models to new tasks or domains.

- **Subspace Similarity and Transferrability Metrics**: We propose a subspace similarity metric to identify pairs of modules between source and target base models that share a common subspace. Additionally, we compute a scalar transferability cost metric to quantify the difficulty of transferring adapters. This metric is derived from the optimal transport solution to pair-wise subspace similarities between different modules of the source and target models.

- **Transfer Method**: We present a practical yet highly effective method to transfer LoRA-X from a source model to a target model. This approach ensures that the target model inherits the capabilities of the source model, enabling efficient knowledge transfer and improved performance.

Throughout this paper, we refer to "Source" as the case where the LoRA-X adapter is trained from scratch on a specific base model (pretrained weights) using a training dataset. Conversely, "Target" denotes the case where the LoRA-X adapter, transferred from a different base model without any additional training, is applied to a new base model.

## 2 Related Work

**Parameter Efficient Finetuning** (PEFT) (Xu et al., 2023) has emerged as a pivotal research area, particularly in transfer learning, where the challenge lies in adapting large pretrained models to specific tasks without extensive retraining. The literature on PEFT includes various strategies, each aiming to modify a minimal number of parameters while maintaining competitive performance. Several PEFT methods have been introduced, such as Adapter Modules (Sung et al., 2022), Prompt Tuning (Lester et al., 2021), and popular Low-Rank Adaptation techniques like LoRA (Hu et al., 2022) and VeRA (Kopiczko et al., 2023). Among these, recent methods such as SVDiff (Han et al., 2023), PiSSA (Meng et al., 2024), SVFT (Lingam et al., 2024) and LoRA-XS (Bałazy et al., 2024) fine-tune the singular values of base model weight matrices. However, SVDiff and its follow-up works primarily aim to reduce the number of parameters during fine-tuning rather than the transferability of the adapter, which is our main objective. A drawback of existing PEFT techniques is their lack of transferability across base models. Our approach addresses this challenge, providing a novel solution for the first time.

**Knowledge Distillation** (Hinton, 2015; Gou et al., 2021; Kim & Rush, 2016; Park et al., 2019; Bui Thi Mai & Lampert, 2019) is a technique where knowledge from a larger, typically more complex model (teacher) is transferred to a smaller, more efficient model (student). Variants of knowledge distillation include Self-Distillation (Zhang et al., 2019; 2021; Zhang & Sabuncu, 2020), where the same model acts as both teacher and student, and Weak-to-Strong Distillation (Bang et al., 2021; Kaplun et al., 2022; Wang et al., 2022), which can help the stronger model avoid overfitting under certain circumstances. While these approaches have shown promise in transferring knowledge between models, they still rely on a training dataset for the distillation process, making them problematic to apply in data-free scenarios.

**Adapter Transfer**: Both (Wang et al., 2024) and (Ran et al., 2023) aim to transfer adapters across different base models. The primary difference between (Wang et al., 2024) and our method lies in their use of synthetic data generated by the source model, along with a small subset of the original dataset, for the transfer process. (Ran et al., 2023) proposed a universal mapper capable of transferring adapters from a source diffusion model to a target model. However, this mapper requires training for each target model using a dataset subset common to both the source and target models. In contrast, we develop an adapter transfer methodology using a closed-form solution, which eliminates the need for additional training. Additionally, we introduce adapters that are inherently transferable.

## 3 Motivation

A LoRA adapter is fine-tuned for a specific task using a designated base model and dataset, making it dependent on that model. This dependency becomes problematic if the base model is updated or if a user wants to use a distilled version, like SDXL to SSD-1B. Additionally, users might lack access to the original training dataset, complicating transfers. To address this, we propose LoRA-X, an adapter that can be easily transferred between different base model versions. This approach leverages the strong correlation between layers of different base model versions (Samragh et al., 2023), especially in deeper layers, which significantly impact fine-tuned task performance (Frenkel et al., 2024).

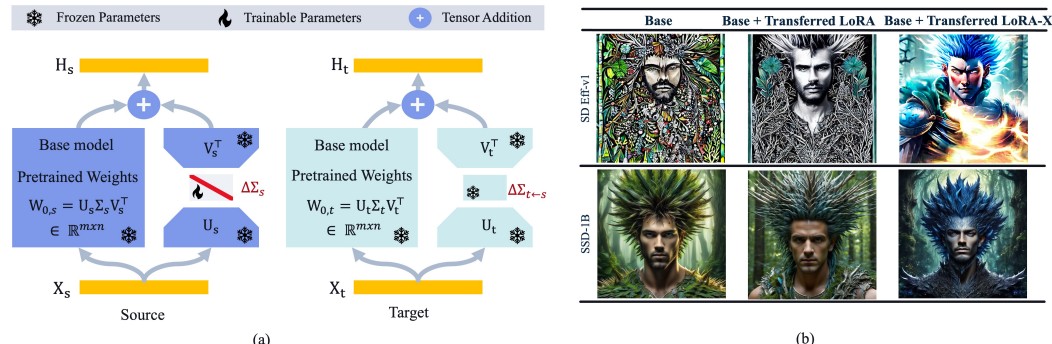

Figure 2: (a) Source: only $\Delta\boldsymbol{\Sigma}_s$ is finetuned for a down stream task, Target: for a given $\Delta\boldsymbol{\Sigma}_s$ from a source, first computes $\Delta\boldsymbol{\Sigma}_{s\leftarrow t}$ and then reconstruct the weight change using its own left and right singular matrices. (b) Samples produced by diffusion target models SD Eff-v1.0 and SSD-1B, utilizing training-free transferred adapters from sources SD-v1.5 and SDXL, respectively.

LoRA-X is designed to stay within the same subspace as the base model, focusing on amplifying or diminishing features relevant to specific tasks without introducing new feature extractors. This method enhances flexibility and adaptability, ensuring fine-tuned tasks benefit from base model updates without losing specific adaptations. Figure2 (b) shows samples generated using a diffusion base model (e.g., SD Eff-v1.0) with various training-free transferred style adapters. Unlike Transferred LoRA, the samples generated using Transferred LoRA-X—transferred from a LoRA-X adapter trained on a different base model (e.g., SD-v1.5) without additional training—successfully capture the BlueFire style.

## 4 METHOD

This section is organized as follows: **LoRA-X**: We introduce LoRA-X and define it based on the Source base model. **Methods for Transferring LoRA-X**: We then explain how to transfer LoRA-X from the Source to the Target model without additional training.

### 4.1 LoRA-X

We design LoRA-X based on the following two criteria: First, (Hu et al., 2022) demonstrated that *LoRA can amplify key features for specific downstream tasks that were learned but not highlighted in the general pre-training model.* Second, we only have access to the target pre-trained model. Therefore, any similarity measurement between different modules of the source and target models can only be based on their pre-trained model weights and their corresponding subspace. Building on these crucial insights, we constrain the adapter to remain within the subspace of the (source) base model. This allows for straightforward transfer to another (target) base model by projecting it into the target's subspace.

To ensure that the adapter $\Delta\boldsymbol{W} \in \mathbb{R}^{m \times n}$ remains in the same subspace as the base model weight $\boldsymbol{W}_0 \in \mathbb{R}^{m \times n}$, $\Delta\boldsymbol{W}$ should adhere to a specific structure. This typically involves ensuring that $\Delta\boldsymbol{W}$ is aligned with the principal components or directions of $\boldsymbol{W}_0$. One common approach is to express $\Delta\boldsymbol{W}$ in terms of the singular value decomposition (SVD) of $\boldsymbol{W}_0$. If $\boldsymbol{W}_0 = \boldsymbol{U}\boldsymbol{\Sigma}\boldsymbol{V}^\top$, where $\boldsymbol{U}$ and $\boldsymbol{V}$ are left and right singular matrices, respectively, and $\boldsymbol{\Sigma} = \mathrm{diag}(\boldsymbol{\sigma})$ and $\boldsymbol{\sigma} = [\sigma_1, \ldots, \sigma_n]$ is a diagonal matrix of singular values in descending order[1], then $\Delta\boldsymbol{W}$ can be structured as:

$$\Delta\boldsymbol{W} = \tilde{\boldsymbol{U}}\Delta\boldsymbol{\Sigma}\tilde{\boldsymbol{V}}^\top \tag{1}$$

Here, $\tilde{\boldsymbol{U}} \in \mathbb{R}^{m \times r}$ and $\tilde{\boldsymbol{V}} \in \mathbb{R}^{n \times r}$ are truncated left and right singular matrices obtained by zeroing out the $n - r$ smallest singular values from $\boldsymbol{U}$ and $\boldsymbol{V}$ respectively. The matrix

---

[1]Note that the SVD computation is performed only once and can be cached.

$\Delta\boldsymbol{\Sigma} = \text{diag}([\delta\sigma_1, \ldots, \delta\sigma_r]) \in \mathbb{R}^{r \times r}$ is a diagonal matrix satisfying $\sigma_i + \delta\sigma_i \geq 0, \forall 1 \leq i \leq r$. The parameter $r$ represents the rank of $\Delta\boldsymbol{W}$ identifying the rank of the adapter. During fine-tuning task, $\tilde{\boldsymbol{U}}$ and $\tilde{\boldsymbol{V}}$ are frozen and only $\Delta\boldsymbol{\Sigma}$ learns the change of the singular values of the pre-trained model.

By projecting $\Delta\boldsymbol{W}$ onto $\boldsymbol{W}$ subspace, we have

$$\boldsymbol{U}\boldsymbol{U}^{\top}\Delta\boldsymbol{W}\boldsymbol{V}\boldsymbol{V}^{\top} = \boldsymbol{U}\boldsymbol{U}^{\top}\tilde{\boldsymbol{U}}\Delta\boldsymbol{\Sigma}\tilde{\boldsymbol{V}}^{\top}\boldsymbol{V}\boldsymbol{V}^{\top} = \tilde{\boldsymbol{U}}\Delta\boldsymbol{\Sigma}\tilde{\boldsymbol{V}}^{\top} = \Delta\boldsymbol{W} \qquad (2)$$

where $\boldsymbol{U}\boldsymbol{U}^{\top}\Delta\boldsymbol{W}\boldsymbol{V}\boldsymbol{V}^{\top}$ gives the "projection" of $\Delta\boldsymbol{W}$ onto the subspace spanned by $\boldsymbol{W}$. As shown in equation 2 by projection we get $\Delta\boldsymbol{W}$ back, indicating $\Delta\boldsymbol{W}$ was already within the $\boldsymbol{W}$ subspace. Figure 2 (a) illustrates LoRA-X training on a source base model, while (b) shows the transfer of LoRA-X into a target model.

Generally, the matrix $\Delta\boldsymbol{\Sigma}$ can be any arbitrary square matrix and does not need to be diagonal. This flexibility allows the adapter to capture a wide range of transformations. However, if the general pre-trained model has already learned the key features necessary for specific downstream tasks, imposing a diagonal constraint on $\Delta\boldsymbol{\Sigma}$ can be sufficient. This is because the diagonal elements can scale the learned features appropriately without needing to mix them. In cases where the pre-trained model's features are not perfectly aligned with the downstream task, the adapter must learn a linear combination of these features. This is achieved by allowing $\Delta\boldsymbol{\Sigma}$ to be a full matrix, enabling the adapter to reweight and combine the pre-trained features in a more complex manner.

(Han et al., 2023) proposed a similar adapter structure called SVDiff, targeting the fine-tuning of singular values within weight matrices. Our method differs in two key ways: we apply truncated SVD, modifying only the $r$ largest singular values, and we apply LoRA-X only to attention modules, while SVDiff applies the adapter to all modules, including convolutional ones. SVDiff aims to minimize the number of parameters adjusted during fine-tuning, which is beneficial in resource-limited scenarios.

In contrast, our work emphasizes the transferability of the adapter, enhancing adaptability across different base models. By focusing on transferability, we aim to create an adapter that can be effectively reused in various LFMs, improving both efficiency and effectiveness. This distinction highlights the different priorities: SVDiff focuses on parameter efficiency, while our approach emphasizes transferability and adaptability.

## 4.2 Methods for Transferring LoRA-X

Assume a source model with the base model weight $\boldsymbol{W}_{s,0} = \boldsymbol{U}_s\boldsymbol{\Sigma}_s\boldsymbol{V}_s^{\top} \in \mathbb{R}^{m \times n}$ and the LoRA-X weight $\Delta\boldsymbol{W}_s = \tilde{\boldsymbol{U}}_s\Delta\boldsymbol{\Sigma}_s\tilde{\boldsymbol{V}}_s^{\top} \in \mathbb{R}^{m \times n}$. Our goal is to transfer $\Delta\boldsymbol{W}_s \in \mathbb{R}^{m \times n}$ to another target model with the base weight $\boldsymbol{W}_{t,0} = \boldsymbol{U}_t\boldsymbol{\Sigma}_t\boldsymbol{V}_t^{\top} \in \mathbb{R}^{m' \times n'}$ without training.

### 4.2.1 Same Dimension

In the case, $m = m'$ and $n = n'$, by projecting the source adapter weight $\Delta\boldsymbol{W}_s$ into the target pre-trained based weight $\boldsymbol{W}_t = \boldsymbol{U}_t\boldsymbol{\Sigma}_t\boldsymbol{V}_t^{\top}$ we have

$$\Delta\boldsymbol{W}_{t \leftarrow s} = \boldsymbol{U}_t\boldsymbol{U}_t^{\top}\Delta\boldsymbol{W}_s\boldsymbol{V}_t\boldsymbol{V}_t^{\top} = \boldsymbol{U}_t\boldsymbol{U}_t^{\top}\tilde{\boldsymbol{U}}_s\Delta\boldsymbol{\Sigma}_s\tilde{\boldsymbol{V}}_s^{\top}\boldsymbol{V}_t\boldsymbol{V}_t^{\top} = \boldsymbol{U}_t\Delta\boldsymbol{\Sigma}_{t \leftarrow s}\boldsymbol{V}_t^{\top} \qquad (3)$$

where $\Delta\boldsymbol{\Sigma}_{t \leftarrow s} = \boldsymbol{U}_t^{\top}\tilde{\boldsymbol{U}}_s\Delta\boldsymbol{\Sigma}_s\tilde{\boldsymbol{V}}_s^{\top}\boldsymbol{V}_t$ rotates $\Delta\boldsymbol{\Sigma}_s$ in a direction that the source and target have highest subspace similarity. Consequently, $\Delta\boldsymbol{\Sigma}_{t \leftarrow s}$ can influence the target's modules that have pre-trained weight subspaces highly similar to those in the source model. Moreover, $\Delta\boldsymbol{\Sigma}_{t \leftarrow s}$ is no longer a diagonal matrix unless the source and target subspaces are perfectly aligned, i.e., $\boldsymbol{U}_t\boldsymbol{U}_t^{\top}\tilde{\boldsymbol{U}}_s = \tilde{\boldsymbol{U}}_t$ and $\tilde{\boldsymbol{V}}_s^{\top}\boldsymbol{V}_t\boldsymbol{V}_t^{\top} = \tilde{\boldsymbol{V}}_t^{\top}$.

### 4.2.2 Different Dimensions

If the dimensions of the source and target base model weights do not match, with either $m \neq m'$ or $n \neq n'$, we cannot compute $\boldsymbol{U}_t^{\top}\tilde{\boldsymbol{U}}_s$ or $\tilde{\boldsymbol{V}}_s^{\top}\boldsymbol{V}_t$ in equation 3. Instead, we need to find a subspace of the same dimension that captures the highest correlation. One possible

solution is to find a linear transformation that minimizes the Frobenius norm difference. Assuming $m \neq m'$, the linear transformation can be evaluated as $\hat{P} = \arg\min_{P} \|PU_s - U_t\|_F^2$. Consequently, we have $\tilde{U}_s = U_t U_s^\top (U_s U_s^\top)^{-1} U_s$. Similarly, if $n \neq n'$, an approximate of right singular matrix is given by $\tilde{V}_s = V_s (V_s^\top V_s)^{-1} V_s^\top V_t$.

### 4.2.3 Subspace Similarity

To capture subspace similarity, we use **unweighted similarity**,

$$\Phi_l(A, B) = \Psi(U_A, U_B) = \frac{\|U_A^\top U_B\|_F^2}{n} = \frac{\sum_i \sum_j \langle u_A^i, u_B^j \rangle^2}{n} \qquad (4)$$

to measure the subspace similarity between two column orthonormal matrices $U_A \in \mathbb{R}^{m \times n}$ and $U_B \in \mathbb{R}^{m \times n}$, obtained by taking columns of the left singular matrices of $A \in \mathbb{R}^{m \times n}$ and $B \in \mathbb{R}^{m \times n}$. Similarly, we use $\Phi_r(A, B) = \Psi(V_A, V_B) = \frac{\|V_A^\top V_B\|_F^2}{n}$ by taking columns of $V_A \in \mathbb{R}^{n \times n}$ and $V_B \in \mathbb{R}^{n \times n}$ the right singular matrices of $A$ and $B$, respectively. Refer to Appendix C for additional variants of the subspace similarity metric.

### 4.2.4 Transferability Metric Across Models

To determine the transferability of adapters across models, we use subspace similarity between modules of a source and target base model. We propose a transferability metric to guide training-free transfers between architectures, such as from SDXL to SD1.5. This involves developing a cost function for transferring adapters between architectures.

Assume that the source model has $S$ locations for attaching adapters, with weight matrices $W_s^i$ for $i \in \{1, 2, ...S\}$. Similarly, the target model has $T$ locations with weight matrices $W_t^j$ for $j \in \{1, 2, ...T\}$. The cost of transferring adapter $i$ from the source to $j$ in the target is computed using subspace similarity $\Phi(W_s^i, W_t^j)$. This similarity is then used to construct a cost matrix. The cost matrix is employed to compute an optimal transport map, resulting in an optimal transport cost, which we refer to as the Adapter Transferability Cost (ATC). The ATC, normalized between 0 and 1, indicates the cost of transferring adapters, with higher values representing greater difficulty. Details on obtaining the ATC are in Appendix D

## 5 Experiment

This section describes our experiments evaluating the effectiveness of LoRA-X, detailing the setup and presenting our evaluation. We analyze and quantify LoRA-X through text-to-image generation experiments in Section 5.1 and text-generation experiments in Appendix E.3.

### 5.1 Experimental Setup for Text-To-Image Generation

To evaluate the quality of images generated by LoRA-X and its training-free transferred version, we consider two scenarios. In the first scenario, "Trained", we train the LoRA-X on a specific base model (e.g., SD Eff-v1.0) using a training dataset and generate samples with this trained LoRA-X. In the second scenario, "Transferred", we transfer a LoRA-X trained on a different base model (e.g., SD-v1.5) to the same base model used in the "Trained" scenario (SD Eff-v1.0) and generate samples with the transferred LoRA-X.

**Datasets:** For style transfer, we evaluate LoRA-X trained from scratch on base source models and training-free transferred LoRA-X using public datasets like *BlueFire*, *Origami Styles*, and *Paintings*. We follow the setup described in (Borse et al., 2024). Additional details are in AppendixB.

**Models:** For text-to-image generation tasks, we employ Stable Diffusion v1.5 (SD-v1.5) (Rombach et al., 2022) and Stable Diffusion XL (SDXL) (Podell et al., 2024) as the source models. The target models include Stable Diffusion Efficient v1.0 (SD Eff-v1.0) (see Appendix A for details), Realistic Vision v3.0 (RealVis-v3.0), Segmind Stable Diffusion 1B (SSD-1B) (Gupta et al., 2024), and Realistic Vision XL v3.0 (RealVisXL-v3.0).

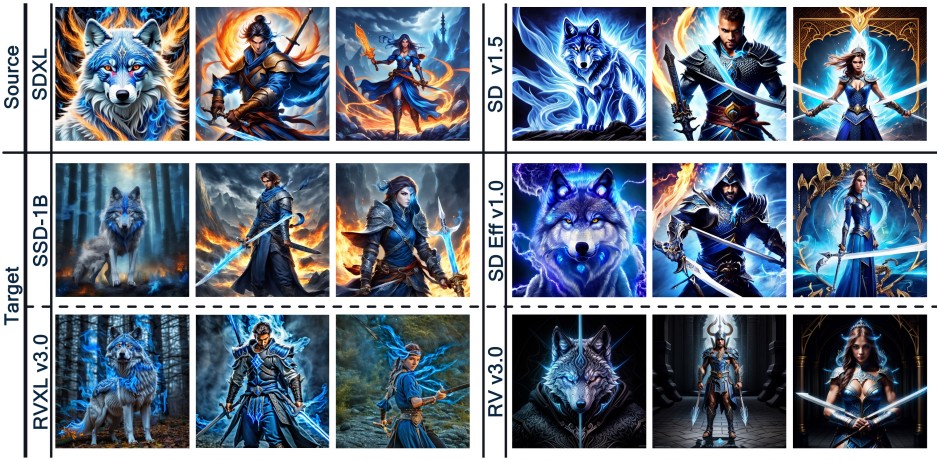

Adapter: "BlueFire", Prompt: 1) "wolf in a forest, long fur" 2) "a man with sword" 3) "a woman with sword"

Figure 3: Generated samples using LoRA-X style adapter on the SDXL and SD-v1.5 as the source base models and corresponding training-free transferred samples using SSD-1B and SD Eff-v1.0 as the target based models.

Table 1: Evaluation of LoRA-X trained from scratch on base models versus training-free transferred LoRA-X from a source model into a target model. LoRA-X modifies the 320 largest singular values of the pre-trained weights. Results are averaged over 30 seeds.

| Datasets | Base Model | Adapter | Training-Free | HPSv2 (↑) | LPIPS diversity (↑) | DINOv2 (↑) |
|---|---|---|---|---|---|---|
| BlueFire (900 images) | RealVis-v3.0 | Trained | | 0.331 | 0.524 | 0.882 |
| | | Transferred | ✓ | 0.332 (+**0.3**%) | 0.540 (+**2.9**%) | |
| | SD Eff-v1.0 | Trained | | 0.296 | 0.534 | 0.851 |
| | | Transferred | ✓ | 0.307 (+**3.6**%) | 0.538 (+**0.7**%) | |
| | RealVisXL-v3.0 | Trained | | 0.319 | 0.484 | 0.947 |
| | | Transferred | ✓ | 0.319 (**0.0**%) | 0.456 (−**6.1**%) | |
| | SSD-1B | Trained | | 0.316 | 0.428 | 0.969 |
| | | Transferred | ✓ | 0.300 (−**5.3**%) | 0.392 (−**8.4**%) | |
| Paintings (630 images) | RealVis-v3.0 | Trained | | 0.319 | 0.502 | 0.928 |
| | | Transferred | ✓ | 0.329 (+**3.0**%) | 0.441 (−**11.8**%) | |
| | SD Eff-v1.0 | Trained | | 0.298 | 0.485 | 0.820 |
| | | Transferred | ✓ | 0.292 (−**2.0**%) | 0.476 (−**2.0**%) | |
| | RealVisXL-v3.0 | Trained | | 0.333 | 0.467 | 0.945 |
| | | Transferred | ✓ | 0.325 (−**2.5**%) | 0.421 (−**9.6**%) | |
| | SSD-1B | Trained | | 0.319 | 0.409 | 0.961 |
| | | Transferred | ✓ | 0.320 (+**0.3**%) | 0.355 (−**13.2**%) | |

**Metrics:** To evaluate image quality in the "Trained" and "Transferred" scenarios, we report DINOv2 (Oquab et al., 2024), HPSv2.1 (Wu et al., 2023), and LPIPS (Zhang et al., 2018) diversity scores. DINOv2 measures similarity based on embedded representations, HPSv2 assesses image quality and prompt/style alignment, and LPIPS captures diversity among generated images across different seeds.

## 5.2 LoRA-X Performance

Table 1 compares the performance of LoRA-X in the "Trained" scenario, where it is trained on various base models using the BlueFire and Painting datasets, with its performance in the "Transferred" scenario, where it is moved from a source model with a different base model to the target model using the same base model as in the "Trained" scenario. Appendix E presents LoRA-X's performance on the Origami dataset.

To enable training-free transfer from a source model to a target model, we begin by identifying the correlated modules between the source and target using Equation equation 4. For additional details, please refer to Appendix C. Subsequently, we project the source's LoRA-X onto the corresponding module in the target model using Equation equation 2. Appendix F provides a PyTorch pseudocode for LoRA-X transfer.

The HPSv2 and LPIPS scores in both scenarios are very similar, indicating that the Transferred LoRA-X performs comparably to the one Trained with the datasets, demonstrating the effective transferability of LoRA-X. Additionally, the high DINOv2 scores suggest that the generated samples in both scenarios are highly correlated.

Figure 3 shows samples generated by LoRA-X based on the BlueFire style dataset. The first row displays samples from the source models SDXL and SD-v1.5 using LoRA-X trained on BlueFire. The second and third rows show samples from training-free transferred LoRA-X to the target models SSD-1B, SD Eff-v1.0, RVXL-v3.0, and RV-v3.0.

## 5.3 EFFECT OF SUBSPACE CONSTRAINT

### 5.3.1 COMPARISON WITH LORA

Table 2 demonstrates the impact of the subspace constraint imposed in the LoRA-X structure (equation 1) by comparing its transferability with LoRA (Hu et al., 2022). For this comparison, we used the BlueFire dataset and fine-tuned both LoRA and LoRA-X on SD-v1.5 as the source model, with SD Eff-v1.0 as the target. This comparison highlights how the subspace constraint affects the adapter's ability to transfer across different base models. In this experiment, we trained the LoRA $\Delta W = BA$ of rank $r = 32$ on SD-v1.5 as the base model. Subsequently, we transferred LoRA to SD Eff-v1.0 by projecting into its pre-trained weights subspace, i.e., $BA_{t \leftarrow s(\text{SD-v1.5})} = U_t U_t^\top BA V_t V_t^\top$. The evaluation demonstrates that subspace constraint is essential for maintaining quality and diversity of transferred style. We repeated the experiment for different LoRA ranks to show how LoRA's transferability drops as rank is reduced, though its total size remains much higher than LoRA-X. Appendix E.1 presents the effect of constraint based on the Origami dataset.

Table 2: LoRA-X subspace constraint effect on transferability of style adapter. BlueFire dataset, SD-v1.5 as the source model and SD Eff-v1.0 as the target.

| Method | Adapter | Rank | HPSv2 (↑) | LPIPS diversity (↑) | DINOv2 (↑) | Total size (MB) |
|--------|---------|------|-----------|---------------------|------------|-----------------|
| LoRA-X | Trained | 320 | 0.2958 | 0.5340 | **0.8513** | 0.16 |
| | Transferred | | 0.3073 (+**3.7**%) | 0.5376 (+**0.6**%) | | |
| LoRA | Trained | 32 | 0.3153 | 0.5049 | 0.8471 | 34.07 |
| | Transferred | | 0.2466 (−**27.8**%) | 0.4834 (−**4.4**%) | | |
| | Trained | 16 | 0.2652 | 0.5248 | 0.8266 | 17.08 |
| | Transferred | | 0.2408 (−**10.1**%) | 0.5224 (−**0.5**%) | | |
| | Trained | 1 | 0.2650 | 0.5312 | 0.8228 | 1.15 |
| | Transferred | | 0.2355 (−**12.5**%) | 0.5274 (−**0.7**%) | | |

### 5.3.2 COMPARISON WITH DORA AND FOURA

Additionally, we showed results for DoRA (Liu et al., 2024) and FouRA (Borse et al., 2024) adapters in Table 3. From the results, we see that the projection idea works well on both these type of adapters. However, the DINO score of the transfer is relatively small compared to that of LoRA-X transfer. Moreover, the percentage change of transferred and trained adapters are higher suggesting that LoRA-X transfers better.

Table 3: Transferability of style adapters DoRA & FouRA. For DoRA, SDXL is the source model and SSD-1B is the target model. For FouRA, SD-v1.5 is the source model and SD Eff-v1.0 is the target model.

| Method | Adapter | Rank | Dataset | HPSv2 (↑) | LPIPS diversity (↑) | DINOv2 (↑) |
|--------|---------|------|---------|-----------|---------------------|------------|
| DoRA | Trained | 8 | Paintings | 0.3042 | 0.4624 | **0.9138** |
| | Transferred | | | 0.2764 (−**9.1**%) | 0.4526 (−**2.1**%) | |
| DoRA | Trained | 8 | Origami | 0.2491 | 0.3408 | **0.9441** |
| | Transferred | | | 0.2224 (−**10.7**%) | 0.3073 (−**9.8**%) | |
| FouRA | Trained | 64 | Paintings | 0.3034 | 0.4686 | **0.9153** |
| | Transferred | | | 0.2891 (−**4.7**%) | 0.4446 (−**5.1**%) | |

## 5.4 Comparison with X-adapter

We compare the performance of transferred LoRA-X using our training-free method based on equation 3 with X-Adapter (Ran et al., 2023), which uses plug-and-play modules trained on the target model. Table 4 shows the comparison: the "Transferred" row for LoRA-X indicates our training-free transfer from SSD-1B to SDXL, while for X-Adapter, it refers to the transfer method using X-adapter modules trained for SD-v1.5 to SDXL. The "Trained" row for both methods refers to trained LoRA-X adapter from scratch using BlueFire dataset.

Table 4: Evaluation of training-free transferred LoRA-X from SSD-1B to SDXL versus LoRA-X trained on SDXL from scratch using BlueFire dataset using our training-free transfer method and training-based X-adapter. Wall clock time is measured on A100 GPU

| Method | Adapter | HPSv2 (↑) | LPIPS diversity (↑) | DINOv2 (↑) | Time (↓) |
|---|---|---|---|---|---|
| LoRA-X | Trained | 0.306 | 0.422 | 0.953 | 3.7s |
|  | Transferred | 0.279 ($-$**9.5**%) | 0.433 ($+$**2.6**%) |  |  |
| X-Adapter | Trained | 0.306 | 0.422 | 0.892 | 17.1 s |
|  | Transferred | 0.282 ($-$**7.8**%) | 0.406 ($-$**3.7**%) |  |  |

Results show change in performance for HPSv2 & LPIPS from the trained baseline is in similar. However, our LoRA-X transfer produces higher DINO score mainly because it is transferred from a source in the similar family i.e SSD-1B. Also inference time for X-adapter is higher due to processing through base model, transferred model and the adapter.

## 5.5 Ablation Studies

### 5.5.1 Effect of Subspace Projection

Table 5 illustrates the impact of subspace projection in LoRA-X by comparing the training-free transfer of LoRA-X from a source model using subspace projection equation 2 with the method of directly copying $\Delta\mathbf{\Sigma}_s$ from the source to the target, i.e., $\Delta\widehat{\boldsymbol{W}}_{t\leftarrow s} = \boldsymbol{U}_t\Delta\boldsymbol{\Sigma}_s\boldsymbol{V}_t^\top$. This analysis focuses on the effect of the alignment of left and right singular matrices between the source and target models. As shown in the table, without subspace projection, the performance of transferred LoRA-X significantly drops, indicating that subspace projection is crucial. Appendix E shows the effect of subspace projection in the SDXL family.

Table 5: Evaluation of training-free transformed LoRA-X by copying singular value modifications from the source to the target versus subspace projected one.

| Subspace Proj. | HPSv2 (↑) | LPIPS diversity (↑) | DINOv2 (↑) |
|---|---|---|---|
|  | 0.1235 | 0.4804 | 0.7046 |
| ✓ | **0.3073** | **0.5376** | **0.8513** |

### 5.5.2 LoRA-X Rank

We analyze LoRA-X performance at various ranks, indicating the number of modified singular values. Table 6 shows that trained LoRA-X (trained on SD Eff-v1.0) performance declines as rank decreases. In contrast, transferred LoRA-X (from SD-v1.5) maintains performance close to the Trained version.

Table 6: LoRA-X subspace constraint effect on transferability of style adapter. BlueFire dataset, SD-v1.5 as the source model and SD Eff-v1.0 as the target.

| Method | Adapter | Rank | HPSv2 (↑) | LPIPS diversity (↑) | DINOv2 (↑) | Total size (MB) |
|---|---|---|---|---|---|---|
| LoRA-X | Trained | 320 | 0.2958 | 0.5340 | 0.8513 | 0.16 |
|  | Transferred |  | 0.3073 ($+$**3.7**%) | 0.5376 ($+$**0.6**%) |  |  |
|  | Trained | 160 | 0.2850 | 0.5310 | 0.8352 | 0.1 |
|  | Transferred |  | 0.2849 ($-$**0.03**%) | 0.5263 ($-$**0.8**%) |  |  |
|  | Trained | 80 | 0.2782 | 0.5294 | 0.8300 | 0.05 |
|  | Transferred |  | 0.2723 ($-$**2.1**%) | 0.5224 ($-$**1.3**%) |  |  |

### 5.5.3 Smaller Source

In previous sections, we examined the transferability of LoRA-X from larger or similarly sized source models to target models. Here, we demonstrate the performance of transfer

from a smaller source model. Table 7 presents the performance of LoRA-X transfer from SD EFF-v1.0 to SD-v1.5. The Trained row indicates LoRA-X performance when trained on SD-v1.5 from scratch using the BlueFire dataset. The Transferred row indicates LoRA-X training-free transfer from the source model SD Eff-v1.0 to the target model SD-v1.5. See Appendix E for evaluation of LoRA-X transfer from smaller SSD-1B to larger SDXL.

Table 7: Evaluation of Transferred LoRA-X from the smaller source (SD Eff-v1.0) to the larger target (SD-v1.5) versus LoRA-X Trained on SD-v1.5 using BlueFire dataset.

| Adapter | HPSv2 score (↑) | LPIPS diversity (↑) | DINOv2 (↑) |
|---|---|---|---|
| Trained | 0.2959 | 0.5386 | 0.8312 |
| Transferred | 0.2834 (−**4.4%**) | 0.5322 (−**1.2%**) | |

## 5.6 Cross Model Transferability Metric

We measure ATC for different pairs of source and target among following models: SD-v1.5, SD Eff-v1.0, RV-v3.0, SDXL, SSD-1B and RVXL-v3.0. The cost is computed for both the left and the right subspace similarities. The results are shown in Figure 4 (a) and (b), respectively. ATC is normalized to be between 0 and 1, with higher values indicating a larger cost of transferring adapters. From the plot, we observe that for both the left and right subspace similarities, ATC among Family 1 (SD-v1.5, SD Eff-v1.0, and RV-v3.0) is less than 0.5. Similarly, ATC among Family 2 (SDXL, SSD-1B, and RVXL-v3.0) is also less than 0.5.

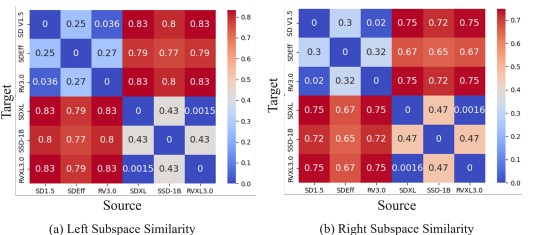

(a) Left Subspace Similarity     (b) Right Subspace Similarity

Figure 4: Adapter transferability cost (ATC) between source and target models using (a) left subspace similarity and (b) right subspace similarity. Lower cost implies easier transfer.

However, ATC across different families shows higher cost, suggesting difficulty in training-free adapter transferability. As expected, ATC is lower between same architectures, such as between SD-v1.5 and RV-v3.0 or SDXL and RVXL-v3.0.

## 5.7 Discussion and Future work

In this paper, we primarily focus on the training-free transferability of LoRA-X in text-to-image generation tasks. To determine transferability, we use subspace similarity between different modules of a source base model and a target base model to assess if LoRA-X is transferable from source to target. We mainly considered architectures from the same family when transferring LoRA. Training-free transfer across different architecture families is a challenge, as demonstrated by our transferability cost. Future research in this area is expected to address these challenges. While our primary focus is on transferring style LoRA-X, future research could extend this to acceleration adapters such as LCM-LoRA (Luo et al., 2023). Additionally, another promising area for future work could involve training-free transfer across different architectures of Large Language Models (LLMs).

## 6 Conclusion

The increasing reliance on LFMs has amplified the need for PEFT methods like LoRA, which offer comparable performance to full model fine-tuning with minimal additional parameters. However, the necessity to retrain LoRA modules when base models are replaced poses significant challenges, especially when the original training data is inaccessible. To address this, we proposed the LoRA-X method, a compact adapter enabling training-free transfer of LoRAs across different base models. This is achieved by maintaining the adapter within the base model's subspace and integrating it into layers with similar subspace characteristics. This innovative approach has been validated with text-to-image generation models, demonstrating its potential to streamline the adaptation process in scenarios where data privacy or availability is a concern.

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

## A    Efficient UNet Architecture

The SD-v1.5 UNet architecture has an attention block in the first three downsampling and the last three upsampling stages. The highest input dimension feature maps to these stages are $64 \times 64$, which are prevalent in the first upsampling stage and the last downsampling stage. Hence, each cross attention block contains linear layers comprising of $4096 \times 4096$ matrices. From an on-device latency standpoint, these blocks incur majority of the computational bottleneck. Furthermore, we observe that much of text and spatial-semantics interaction is captured in low-resolution stages of UNet. These consist of $32 \times 32$ or lesser dimensional feature vectors, along with a higher number of channels and capacity, critical for Diffusion based Generative Models. Hence, we distill our pruned UNet, "SD Efficient-v1", presented in Figure 5, without these $4096 \times 4096$ dimensional cross-attention blocks.

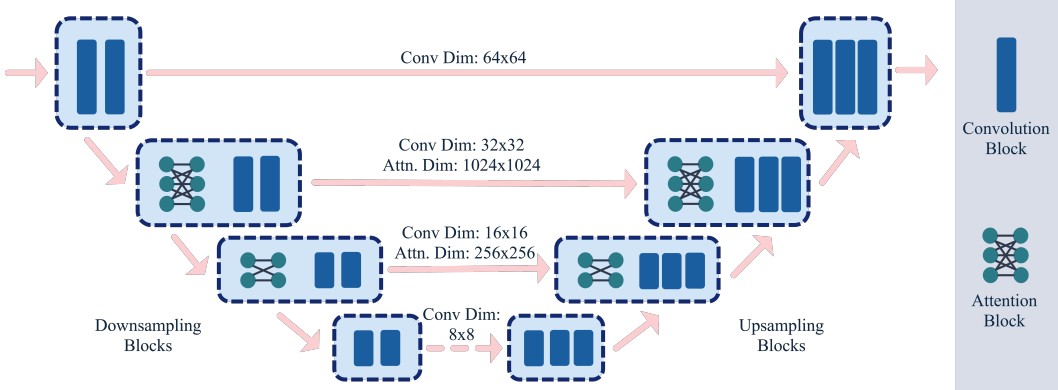

Figure 5: Our Efficient UNet architecture.

## B    Datasets

In this section, we provide more details on the style transfer datasets we use for vision adaptation experiments. We followed the licensing terms for every dataset which was curated.

**BlueFire (Training):** The *BlueFire* dataset is created by collecting images from open public domain and consist of 6 concepts - car, dragon, bird, fox, man and castle. The dataset has a total of 54 images covering all the concepts.

**BlueFire (Validation):** The *Bluefire* validation set consists of 30 curated text prompts, of which 9 prompts contain one of 6 categories on which the model was trained, and the remaining 21 prompts correspond to categories which the low-rank adapter has not been fine-tuned on. These contain categories such as: (football, monster, sword, chess rook, lion, tiger, dog, cat, koala, panda).

For all training experiments validating on this dataset, we produce 30 images per prompt, varying the input seed. Hence, the HPS analysis is over 900 image and LPIPS-diversity analysis is over  14500 image pairs.

**Paintings:** On similar lines, the *Paintings* dataset is also a collection of images from public domain (CC0 license). The dataset has a total of 90 images cover 9 concepts - fire, bird, elephants, ship, horse, flower, woman, man and tiger.

**Paintings (Validation):** The *Paintings* validation set consists of 21 curated text prompts, of which 9 prompts contain one of 9 categories on which the model was trained, and the remaining 12 prompts correspond to categories which the low-rank adapter has not been fine-tuned on. These contain categories such as: (lion, tiger, dog, cat, koala, panda, and other landscapes)

**Origami:** The *Origami* dataset is also a collection of origami images from public domains. The dataset has a total of 52 images covering 7 concepts - bird, boat, flower, cat, dog, fox and house.

**Pokemon:** The *Pokemon* dataset is a collect of Pokemon images, initially introduced in (Liu et al., 2020). It consists of 833 images and captions, the captions have been labeled using BLIP (Li et al., 2022).

### B.1 LoRA-X on SDXL Family

To fine-tune SDXL for a specific downstream tasks, we applied LoRA-X exclusively to the UNet attention processors and the [`to_q`, `to_v`, `to_k`, `to_out`] modules. The rank of the $\Delta\mathbf{\Sigma}_s$ is r=320, i.e., $\Delta\mathbf{W}_s = \tilde{\mathbf{U}}_s \Delta\mathbf{\Sigma}_s \tilde{\mathbf{V}}_s^\top$, $\tilde{\mathbf{U}}_s \in \mathbb{R}^{m \times r}, \tilde{\mathbf{V}}_s \in \mathbb{R}^{n \times r}$, meaning the total number of parameters we fine-tune per module is only 320, which is significantly lower than LoRA (Hu et al., 2022). For the adaptation, we use mixed precision training of FP16, a batch size of 8, gradient accumulation steps of 1, use gradient checkpointing and learning rate of 1e-3 with a constant scheduler. We use SNR Gamma = 5.0 and train for 5000 iterations. For training LoRA-X on SSD-1B and RealVisXL-V3.0, as well as for all the datasets: BlueFire, Paintings and Origami, we follow the same set of hyper-parameters. The implementation is derived from the codebase[2], where the hyper-parameters are described as above.

#### B.1.1 LoRA-X Transfer to SSD-1B

To transfer LoRA-X trained on a source model to a target model, we begin by identifying correlated modules between the source and target models through subspace similarity across their various modules.

Figure 6 (a) illustrates the subspace similarity between the common attention blocks of down-blocks in SDXL and SSD-1B. It shows that certain attention blocks in down- and up-blocks exhibit low subspace similarity. For instance, transformer block 3 (tb.3) in down-block 2 (db.2) has a similarity of less than 0.4, which is significantly lower than other blocks.

To address this, we seek another transformer block within the same module that may have higher similarity. As shown in Figure 6 (b), db.2.attentions.0.tb.3 of SSD-1B exhibits higher similarity with db.2.attentions.0.tb.6 of SDXL. Therefore, we apply LoRA-X of that SDXL block on SSD-1B. We observed the similar behavior in db.2.attentions.1.tb.3 and proceeded accordingly.

Finally, as shown in Figure 7, up.0.attentions.2.tb.4 to tb.7 do not exhibit this behavior, and hence we do not apply LoRA-X of SDXL on those blocks of SSD-1B. For up.0.attentions.0.tb.3 and up.0.attentions.1.tb.3, we proceeded to seek another transformer block using the process described in Figure 6 (b).

After identifying correlated modules, we transfer the LoRA-X of a specific module from the source model to the target model by projecting the LoRA-X onto its pre-trained weight using equation 2.

#### B.1.2 LoRA-X Transfer to RealVisXL-v3.0

To transfer LoRA-X trained on SDXL into RealVisXL-v3.0, we followed the same steps as outlined in section B.1.1. First, we identified the correlated attention blocks, and then we projected LoRA-X into the RealVis-v3.0 subspace.

### B.2 LoRA-X on SD-v1.5 Family

To train LoRA-X on SD-v1.5 for a specific downstream tasks, we applied LoRA-X to the UNet and the text-encoder attention processors and the [`to_q`, `to_v`, `to_k`, `to_out`] modules. The rank of the $\Delta\mathbf{\Sigma}_s$ is r=320.

---

[2]https://shorturl.at/x56s8

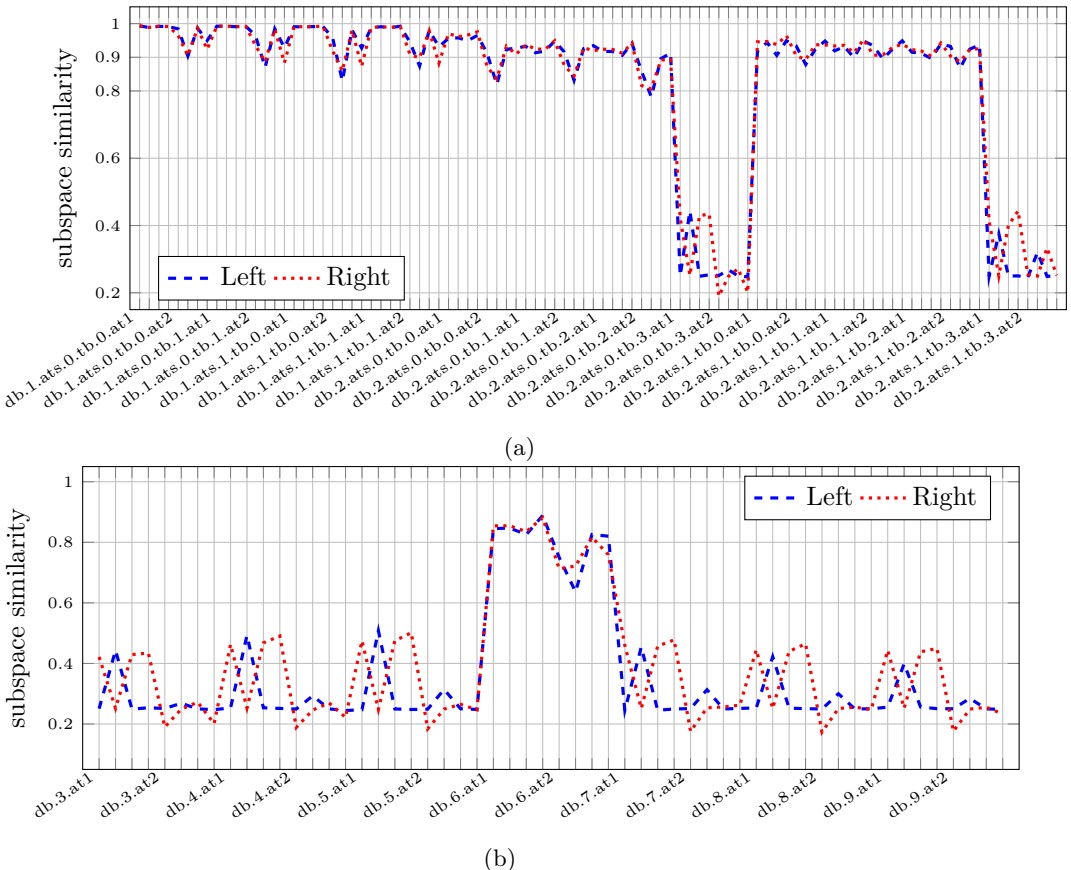

Figure 6: (a) Subspace similarity of common attention blocks of down-blocks (db) in SDXL and SSD-1B, (b) Subspace similarity between db.2.attentions.0.tb.3 of SSD-1B and db.2.attentions.0.tb.3 to db.2.attentions.0.tb.9 of SDXL.

For the adaptation, we use mixed precision training of FP16, a batch size of 8, gradient accumulation steps of 1 and learning rate of 1e-4 with a cosine scheduler. We train for 5000 steps. For training LoRA-X on SD Eff-v1.0 and RealVis-V3.0, as well as for all the datasets: BlueFire, Paintings and Origami, we follow the same set of hyper-parameters.

Table 8 presents the ablation study on hyperparameters, including Steps and Batch size, which differ from the default values in the kohya-ss repository[3] for LoRA finetuing.

Table 8: Ablation on different hyper parameters on training LoRA-X using base model SD-v1.5 and BlueFire dataset.

| Steps | Batch size | HPSv2 ($\uparrow$) | LPIPS ($\uparrow$) |
|-------|------------|--------------------|--------------------|
| 5000  | 4          | 0.284              | 0.528              |
| 2000  | 4          | 0.260              | 0.518              |
| 2000  | 8          | 0.266              | 0.517              |
| **5000** | **8**   | **0.296**          | **0.539**          |

### B.2.1 LoRA-X Transfer

We followed the procedures outlined in section B.1.1 to transfer LoRA-X trained on SD-v1.5 into both SD Eff-v1.0 and RealVis-v3.0.

---

[3]https://github.com/kohya-ss/sd-scripts

To analyze subspace similarity, we followed the same as in section B.1.1. The detailed analysis has been shown in Figure 8 and Figure 9. In fact, the correlation between different subspaces are quite high i.e. greater than 0.8 suggesting direct transfer of the LoRA-X's from the source model to the corresponding target model.

## C  SUBSPACE SIMILARITY

**Weighted similarity**, one can use

$$\Phi_l(A, B) = \frac{\|\boldsymbol{A}^\top \boldsymbol{B}\|_F^2}{\|\boldsymbol{A}^\top \boldsymbol{A}\|_F \|\boldsymbol{B}^\top \boldsymbol{B}\|_F} = \frac{\sum_i \sum_j \sigma_A^i \sigma_B^j \langle \boldsymbol{u}_A^i, \boldsymbol{u}_B^j \rangle^2}{\sqrt{\sum_i (\sigma_A^i)^2} \sqrt{\sum_i (\sigma_B^i)^2}} \tag{5}$$

to measure the subspace similarity between two subspaces spanned by the columns of $\boldsymbol{A}$ and $\boldsymbol{B}$. In this scenario, similarity is influenced by singular values, with larger singular values contributing more significantly to subspace similarity. However, because the adapter can enhance features associated with very small singular values, this similarity measure might not be particularly effective for transferring LoRA-X from a source model to a target model.

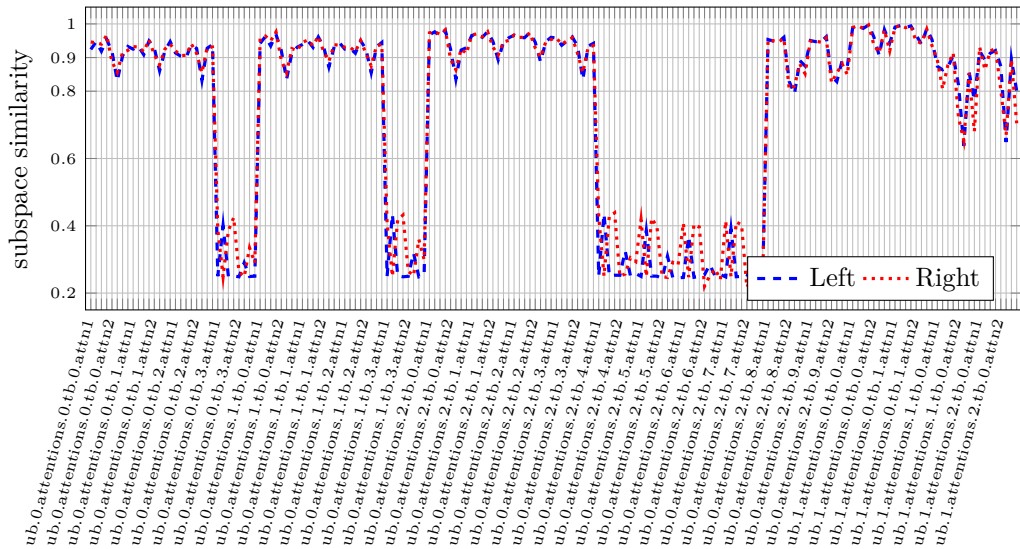

Figure 7: Subspace similarity of common attention blocks in up-blocks (ub) of SDXL vs. SSD-1B.

## D  OPTIMAL TRANSPORT SOLUTION

In this section, we describe the method to compute the optimal transport cost. We calculate $\Phi(\boldsymbol{W}_s^i, \boldsymbol{W}_t^j)$ for all $i \in 1, 2, ...S$ and $j \in 1, 2, ...T$, creating a cost matrix $C \in \mathbb{R}^{S \times T}$, where $C_{ij} = 1 - \Phi(\boldsymbol{W}_s^i, \boldsymbol{W}_t^j)$. Subspace similarity is invalid if: (a) $\boldsymbol{W}_s^i$ and $\boldsymbol{W}_t^j$ have different row and column counts, making it impossible to minimize the Frobenius Norm difference. (b) $\boldsymbol{W}_s^i$ and $\boldsymbol{W}_t^j$ belong to different network parts (e.g., up, down, mid block of UNet) or different attention operations (e.g., query, key, value, output). In such invalid cases, subspace similarity $\Phi(\cdot)$ is considered 0.

This allows us to frame the adapter transfer problem as an optimal transport problem. We solve for the transport map $X \in \mathbb{R}^{S \times T}$ by minimizing:

$$\underset{X}{\text{minimize}}\, \text{Tr}(C^\top X) \quad \text{such that} \quad Xa = \frac{1}{S}b, \quad X^\top b = \frac{1}{T}a, \quad X \geq 0$$

where $a$ is a $T \times 1$-dimensional vector of ones and $b$ is a $S \times 1$-dimensional vector of ones. The standard solution to this problem is using a simplex algorithm, which is inherently

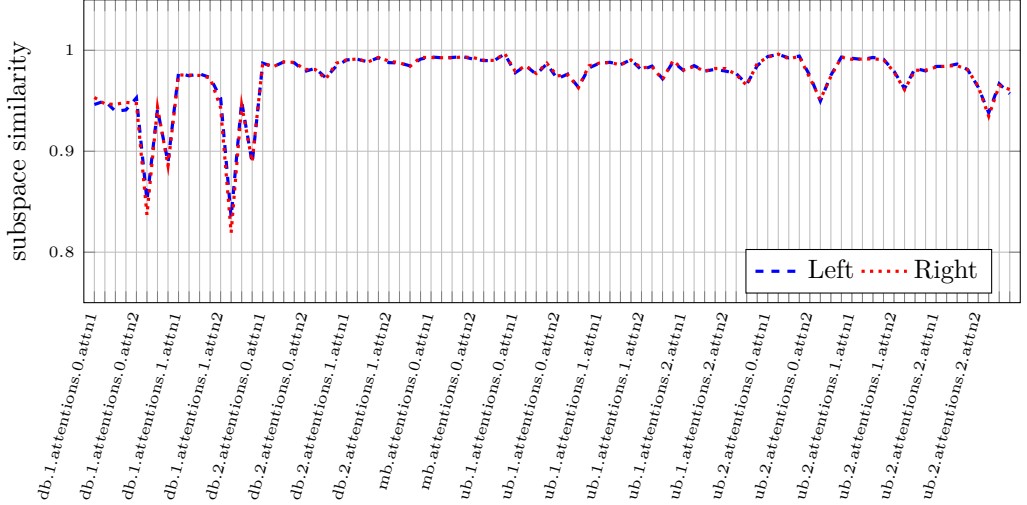

Figure 8: Subspace similarity of common attention blocks in SD-v1.5 vs. SD Eff-v1.

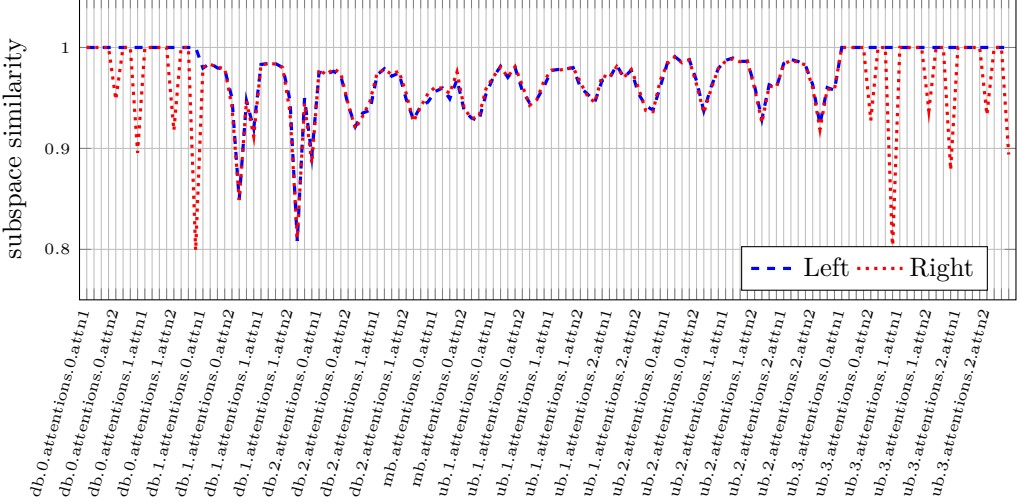

Figure 9: Subspace similarity of common attention blocks in SD-v1.5 and RV-v3.0.

slow. Alternately, we use the network simplex approach which is a graph-theoretic version of the simplex approach. In this version, the linear program is converted into a min-cost flow problem with a bipartite directed graph:

- There are two sets of nodes: source and target. The source has $S$ nodes while the target has $T$ nodes, corresponding to the number of source and target adapter locations.

- The direction of flow is from the source node to the target node, where the supply at each source node is $\frac{1}{S}$ while demand at target node is $\frac{1}{T}$.

- There is a cost associated with flow transfer from $i^{th}$ source node to the $j^{th}$ target node, which is denoted as $C_{ij}$.

Using this setup, we vectorize $X$ into $x$ and $C$ into $c$ and convert the optimization problem to that of

$$\underset{x}{\text{minimize}} \quad c^\top x \quad \text{such that} \quad Ix = d, \quad x \geq 0, \tag{6}$$

where $I \in \mathbb{R}^{(S+T) \times ST}$ is an incidence matrix of the bipartite graph. The rows of the incidence matrix represent the nodes in the graph while the columns represent the edges in the graph. $d \in \mathbb{R}^{(S+T) \times 1}$ is the demand vector where $d_k = -\frac{1}{S}$ for the source nodes and $d_k = \frac{1}{T}$ for the target nodes for $k = \{1, 2, 3, ...S+T\}$. The optimal solution is a basic feasible solution to the problem and can be obtained as a spanning tree of the bipartite graph. The initial solution starts with a spanning tree followed by pivoting to another spanning tree until the one with minimal cost is obtained.

## E   MORE RESULTS

We also evaluate our proposed LoRA-X on the the Origami dataset. On this dataset, we show results of transferring from SDv1.5 to RealVis-v3.0 or SD Eff-v1.0 and from SDXL to RealVisXL-3.0 or SSD-1B in Table 9. The HPSv2 score for both the transferred LoRA-X and that trained from scratch is similar in score. In fact, transferred LoRA-X produces higher performance in most of the cases. This can be attributed to better text-image alignment of the source model from which LoRA-X is transferred. In terms of diversity, the transferred LoRA-X mostly produces poorer performance. This can be due to the fact that all modes of the dataset are not transferred during the subspace projection. However, DINO scores among the dataset generated using transferred LoRA-X and that trained from scratch is high. This suggests that the generated data using both the methods are well correlated.

Table 9: Evaluation of LoRA-X trained from scratch on base source models ($\Delta \Sigma_s$) versus training-free transferred LoRA-X from a source base model into a target model ($\Delta \Sigma_{t \leftarrow s}$) in text-to-image tasks. LoRA-X modifies the 320 largest singular values of the pre-trained weights. Results are averaged over 30 seeds.

| Datasets | Base Model | Adapter | Training-Free | HPSv2 ($\uparrow$) | LPIPS diversity ($\uparrow$) | DINOv2 ($\uparrow$) |
|---|---|---|---|---|---|---|
| Origami (900 images) | RealVis-v3.0 | Trained | | 0.3344 | 0.4476 | 0.9193 |
| | | Transferred | $\checkmark$ | 0.3294 (+**1.5**%) | 0.4497 (+**0.5**%) | |
| | SD Eff-v1.0 | Trained | | 0.2645 | 0.5210 | 0.8184 |
| | | Transferred | $\checkmark$ | 0.2703 (+**2.1**%) | 0.4838 (−**7.7**%) | |
| | RealVisXL-v3.0 | Trained | | 0.2450 | 0.4522 | 0.8847 |
| | | Transferred | $\checkmark$ | 0.2724 (+**9.7**%) | 0.4180 (−**8.1**%) | |
| | SSD-1B | Trained | | 0.2437 | 0.4124 | 0.9413 |
| | | Transferred | $\checkmark$ | 0.2695 (+**9.5**%) | 0.3880 (−**6.3**%) | |

Furthermore, we show visual results on the Painting and Origami datasets in Figure 10 and Figure 11, Figure 12 and Figure 13, respectively, with the SDXL and SD-v1.5 families. Results are shown for the source models as well as cases of transferring LoRA-X to the target model. Visually, we do not see much difference between the generated image styles and contents of both the source or the target models.

### E.1   EFFECT OF SUBSPACE CONSTRAINT

Here we show the effect of subspace constraint using Origami dataset. Table 10 demonstrates the impact of the subspace constraint imposed in the LoRA-X structure (equation 1) by comparing its transferability with LoRA (Hu et al., 2022). For this comparison, we used the Oriami dataset and fine-tuned both LoRA and LoRA-X on SD-v1.5 as the source model, with SD Eff-v1.0 as the target.

### E.2   ABLATION STUDIES ON SDXL FAMILY

In Table 11, we show quantitative results when transferring LoRA-X from SSD-1B to SDXL i.e. from a smaller source to a larger target. The HPSv2 scores and LPIPS diversity scores are quite similar with that of SDXL LoRA-X trained form scratch. This suggests that our LoRA-X is effective even for smaller sources in the SDXL family. Furthermore, high DINO score suggests that image fidelity is quite high between the images generated from both the models.

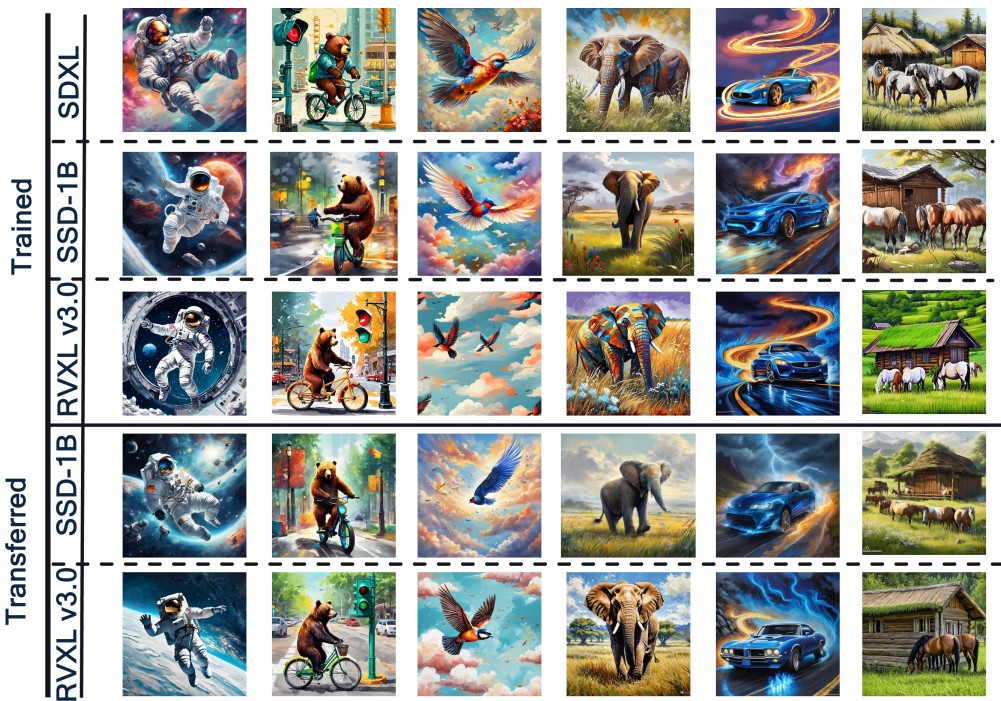

Adapter: "Painting", Prompt: 1) "astronaut floating in space" 2) "bear riding bike, traffic light" 3) "bird flying in the sky" 4) "elephant in a grassland" 5) "car on a winding road, mean headlights, thunderstorms, blue flames" 6) "horses eating grass, wooden hut".

Figure 10: Generated samples using LoRA-X style adapter for painting style on the SDXL as source model and our proposed training-free transfer to SSD-1B and RVXL v3.0. Results are also shown when SSD-1B and RVXL v3.0 are trained from scratch.

Table 10: LoRA-X subspace constraint effect on transferability of style adapter. SD-v1.5 as the source model and SD Eff-v1.0 as the target.

| Dataset | Method | Adapter | Rank | HPSv2 (↑) | LPIPS diversity (↑) | DINOv2 (↑) | Total size (MB) |
|---|---|---|---|---|---|---|---|
| Origami | LoRA-X | Trained
Transferred | 320 | 0.265
0.330 (+**1.5**%) | 0.521
0.484 (−**7.7**%) | **0.819** | 0.16 |
| | LoRA | Trained
Transferred | 32 | 0.253
0.226 (−**10.6**%) | 0.414
0.482 (+**16.4**%) | 0.812 | 34.07 |
| | | Trained
Transferred | 16 | 0.261
0.229 (−**12.2**%) | 0.460
0.475 (+**3.2**%) | 0.781 | 17.08 |
| | | Trained
Transferred | 1 | 0.255
0.230 (−**9.4**%) | 0.480
0.492 (+**2.5**%) | 0.798 | 1.15 |

Table 11: Evaluation of training-free transferred LoRA-X from the smaller source (SSD-1B) to the larger target (SDXL) versus LoRA-X trained on SDXL from scratch using BlueFire dataset.

| Adapter | HPSv2 (↑) | LPIPS diversity (↑) | DINOv2 (↑) |
|---|---|---|---|
| Trained | 0.3060 | 0.4216 | 0.9528 |
| Transferred | 0.2793 (**9.5**%) | 0.4329 (**2.6**%) | |

In Table 12, we show that copying the singular values from source model (SDXL) to target model (SSD-1B) produces slightly poorer performance compared to LoRA-X.

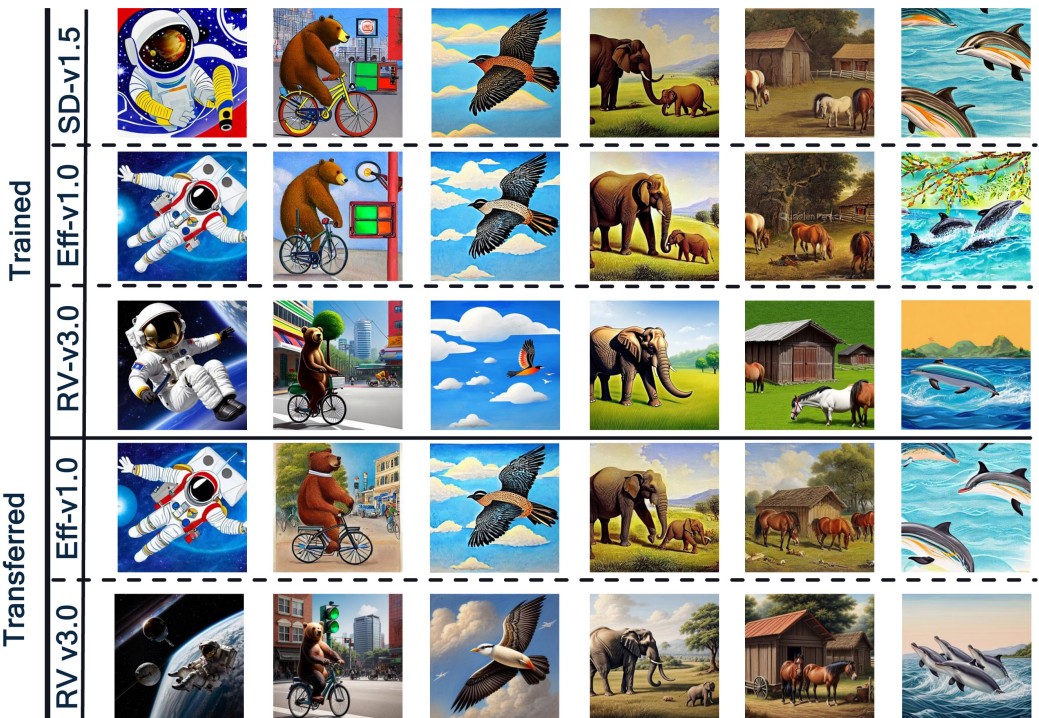

**Adapter:** "Painting", **Prompt:** 1) "astronaut floating in space" 2) "bear riding bike, traffic light" 3) "bird flying in the sky" 4) "elephant in a grassland" 5) "horses eating grass, wooden hut" 6) "wild dolphins swimming".

Figure 11: Generated samples using LoRA-X style adapter for painting style on the SD-v1.5 as source model and our proposed training-free transfer to SD Eff-v1.0 and RV-v3.0. Results are also shown when SD Eff-v1.0 and RV-v3.0 are trained from scratch.

Table 12: Evaluation of training-free transformed LoRA-X by copying singular value modifications from the source (SDXL) to the target (SSD-1B) versus subspace projected one on the BlueFire Dataset.

| Subspace Proj. | HPSv2 (↑) | LPIPS diversity (↑) | DINOv2 (↑) |
|:---:|:---:|:---:|:---:|
| | 0.296 | 0.351 | 0.966 |
| ✓ | 0.300 | 0.392 | 0.969 |

### E.3 Experimental Setup for Text Generation

We have implemented a LoRA-X application to fine-tune TinyLlama, a large language model Zhang et al. (2024), and successfully transferred it from TinyLlama 3T to TinyLlama 2.5T for a prompt generation task using the 'awesome chatgpt prompts' dataset. In this experiment, the rank of LoRA-X is set to $r = 32$. We used the default hyperparameters from the PEFT repository (Mangrulkar et al., 2022) for LoRA finetuning on the CAUSAL task over 3 epochs. Table 13 shows the results for the Trained case versus the Transferred case, indicating that Transferred LoRA-X outperforms in terms of BLEU and ROUGE metrics.

To further assess the generalizability of our approach, we evaluated LoRA-X transferability on two standard text generation benchmarks used in the original LoRA paper (Hu et al., 2022): text-to-text generation on the restaurant review dataset (E2E NLG) (Novikova et al., 2017) (Table 14) and text summarization on the SamSum dataset (Gliwa et al., 2019) (Table 15). In both tasks, we observed only minor differences in BLEU and ROUGE scores between LoRA-X adaptations trained from scratch on the target model versus those trans-

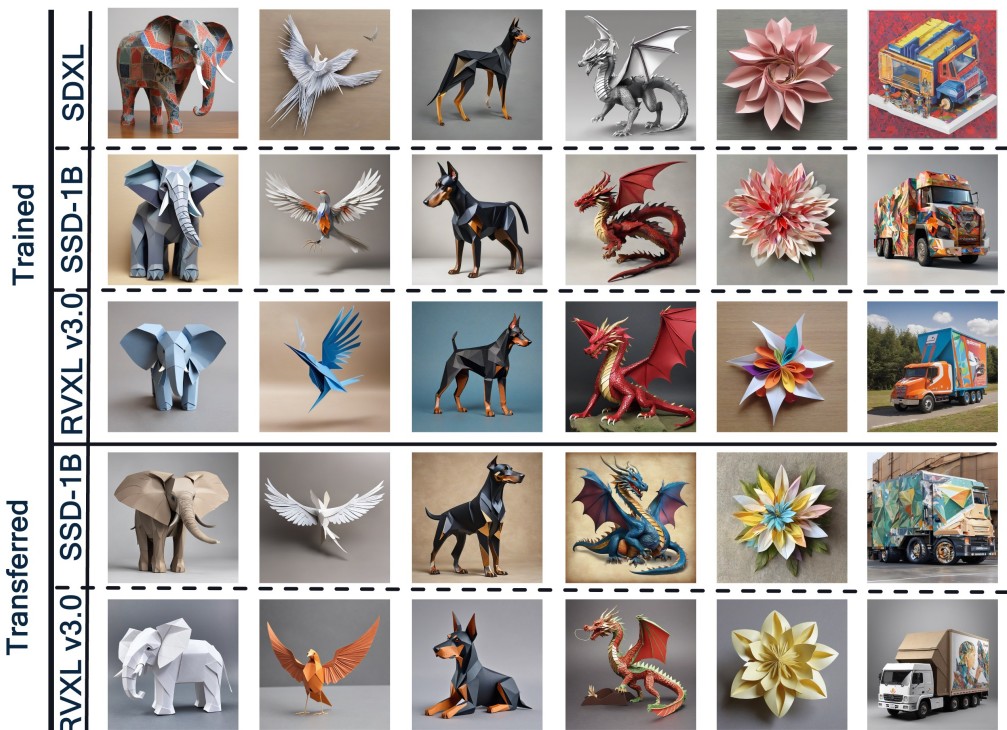

**Adapter: "Origami", Prompt:** 1) "elephant" 2) "bird with spread wings" 3) "doberman dog" 4) "dragon" 5) "flower" 6) "truck".

Figure 12: Generated samples using LoRA-X style adapter for origami style on the SDXL as the source model and our proposed training-free transfer to SSD-1B and RVXL v3.0. Results are also shown when adapters on SSD-1B and RVXL v3.0 are trained from scratch.

ferred from the source model. These results demonstrate the potential of LoRA-X for efficient knowledge transfer across different language tasks and model variants.

Table 13: Evaluation of LoRA-X trained from scratch on the base model TinyLlama 2.5T versus training-free transferred LoRA-X from another base model TinyLlama 3T to the target model TinyLlama 2.5T in a text-generation task using the "awesome chatgpt prompts" dataset.

| Method | Adapter | Bleu (↑) | ROUGE-1 (↑) | ROUGE-2 (↑) | ROUGE-L (↑) | ROUGE-LSum (↑) |
|--------|---------|----------|-------------|-------------|-------------|----------------|
| LoRA-X | Trained | 0.8612 | 0.9349 | 0.9346 | 0.9349 | 0.9349 |
|        | Transferred | 0.8819 | 0.9874 | 0.9873 | 0.9874 | 0.9874 |

Table 14: Evaluation of LoRA-X trained from scratch on the base model TinyLlama 2.5T versus training-free transferred LoRA-X from another base model TinyLlama 3T to the target model TinyLlama 2.5T in a text-generation task using the E2E-NLG dataset.

| Method | Adapter | Bleu (↑) | ROUGE-1 (↑) | ROUGE-2 (↑) | ROUGE-L (↑) | ROUGE-LSum (↑) |
|--------|---------|----------|-------------|-------------|-------------|----------------|
| LoRA-X | Trained | 0.6503 | 0.7689 | 0.6267 | 0.7533 | 0.7533 |
|        | Transferred | 0.6603 | 0.7661 | 0.6423 | 0.7624 | 0.7621 |

## F  IMPLEMENTATION

We also share the implementation of our LoRA-X transfer technique. It takes the LoRA-X from the source model, the target model and filter blocks. The filter blocks are modules

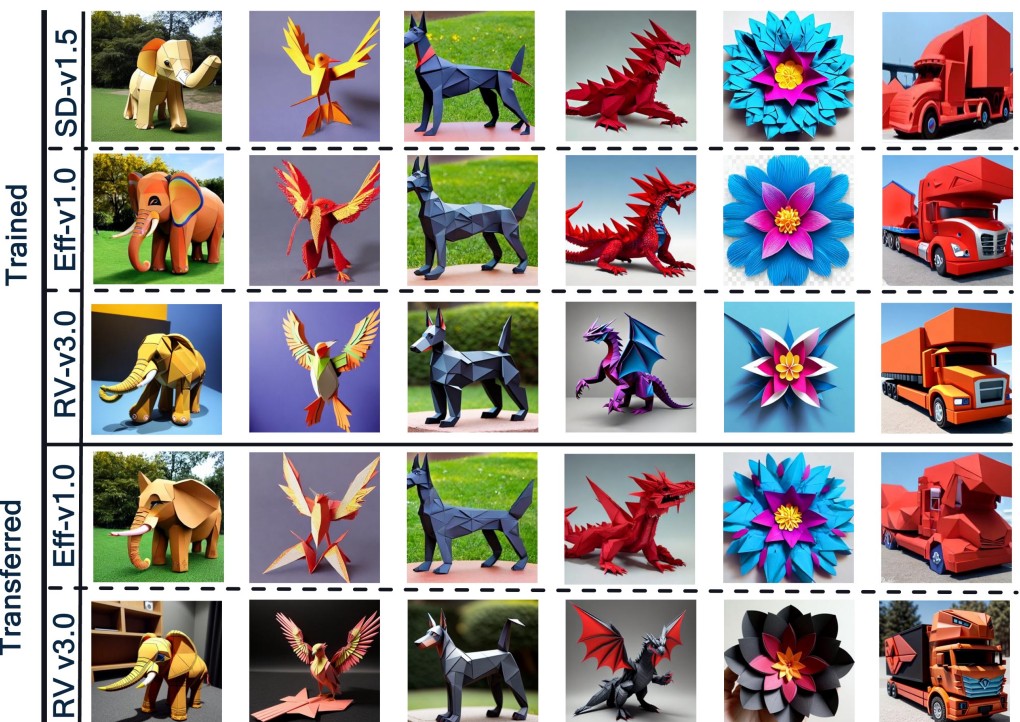

Adapter: "Origami", Prompt: 1) "elephant" 2) "bird with spread wings" 3) "doberman dog" 4) "dragon" 5) "flower" 6) "truck".

Figure 13: Generated samples using LoRA-X style adapter for origami style on the SD-v1.5 as the source model and our proposed training-free transfer to SD Eff-v1.0 and RV-v3.0. Results are also shown when adapters on SD Eff-v1.0 and RV-v3.0 are trained from scratch.

Table 15: Evaluation of LoRA-X trained from scratch on the base model TinyLlama 2.5T versus training-free transferred LoRA-X from another base model TinyLlama 3T to the target model TinyLlama 2.5T in a text-generation task using the SamSum dataset.

| Method | Adapter | ROUGE-1 ($\uparrow$) | ROUGE-2 ($\uparrow$) | ROUGE-L ($\uparrow$) | ROUGE-LSum ($\uparrow$) |
|--------|---------|---------|---------|---------|-----------|
| LoRA-X | Trained | 0.3394 | 0.1394 | 0.2731 | 0.2731 |
|        | Transferred | 0.3568 | 0.1526 | 0.2884 | 0.2882 |

where we do not apply the transfer due to low subspace similarity between the source and target model. The output is the LoRA-X for the target model.

---

**Algorithm 1** Simplistic Pytorch style pseudocode for LoRA-X transfer

---

```python
def forward(ref_lora, tar_model_tensors, filter_blocks):

    '''
    ref_lora: LoRA weights of the reference/source model.
    For LoRA-X case, singular matrix is absorbed into down matrix
    tar_model_tensors: weights of the target model
    filter_blocks: model weights which have low cross similarity
    '''

    tar_lora = {}

    for key, model_key in zip(ref_lora.keys(), tar_model_tensors.keys()):
        # Consider tensors not in the filter_blocks
        if not key.startswith(tuple(filter_blocks)):
            if key.endswith("down.weight"):
                continue

            tar_model_w = tar_model_tensors[model_key]

            lora_up_w = ref_lora[key]
            lora_down_key = key.replace('_lora.up', '_lora.down')

            lora_down_w = ref_lora[lora_down_key]
            lora_rank = lora_down_weight.shape[0]

            u_model_w, s_model_w, vh_model_w = \
                torch.linalg.svd(tar_model_w, full_matrices=False)

            # to project LoRA weights on base model weight
            proj_lora_up_w = u_model_w @ u_model_w.T @ lora_up_w

            proj_lora_down_w = lora_down_w @ vh_model_w.T @ vh_model_w

            tar_lora[key] = proj_lora_up_w
            tar_lora[lora_down_key] = proj_lora_down_w

    return tar_lora
```

---

