# OpenReview forum: "LoRA-X: Bridging Foundation Models with Training-Free Cross-Model Adaptation"
_ICLR.cc/2025/Conference — ICLR 2025 Poster_

### Official Review · Reviewer_7APq · 2024-10-29

**Soundness:** 2
**Presentation:** 2
**Contribution:** 2
**Rating:** 3
**Confidence:** 2

**Summary:**

The paper titled "LoRA-X: Bridging Foundation Models with Training-Free Cross-Model Adaptation" introduces a novel adapter, LoRA-X (Cross-Model Low-Rank Adaptation), which enables the transfer of fine-tuning parameters across different base models without the need for additional training or access to original training data. This is particularly useful when base models are updated or replaced, and retraining adapters is required.

**Strengths:**

1. The work presents LoRA-X, an innovative adapter that enables training-free parameter-efficient transfer across different base models. This approach holds significance in addressing the migration of adapters when base models are updated or replaced, especially considering scenarios where data privacy and licensing issues prevent access to original training data.
2. In introducing the LoRA-X method, the paper provides a solid theoretical analysis and experimental validation. The theoretical part establishes a strong foundation for the design of LoRA-X by comparing the expressiveness of different PEFT (Parameter-Efficient Fine-Tuning) methods. The experimental part verifies the effectiveness of LoRA-X in text-to-image generation tasks, including style transfer and knowledge distillation scenarios. These results support the potential of your method in practical applications. Additionally, the paper offers a detailed analysis of the LoRA-X transfer process, including subspace similarity metrics, which add depth and persuasiveness to the paper.
3. The structure of the paper is clear, and the logic is coherent. From the introduction to related work, and then to the detailed introduction of LoRA-X and experimental results, each part is closely connected and easy to understand.

**Weaknesses:**

1. The paper's comparison of LoRA-X with other parameter-efficient fine-tuning methods is not comprehensive enough. I suggest the authors enhance the comparative analysis with existing methods, particularly the latest ones, to highlight the advantages of LoRA-X and potential areas for improvement.
2. The paper introduces the cross-model adapter LoRA-X but does not emphasize which specific task the method focuses on. While the experimental section shows excellent performance in text-to-image generation tasks, its potential application in other domains is not fully explored. If the method is limited to a particular task, it would be beneficial to clarify this at the beginning of the paper to help readers understand better. Alternatively, if LoRA-X can be applied to multiple different application areas and tasks, a thorough discussion of its performance across various tasks would be valuable.
3. The paper's description of the implementation details of LoRA-X, including the specific implementation and optimization strategies of the algorithm, is not detailed enough. I recommend the authors provide more implementation details, including pseudocode or flowcharts of the algorithm, as well as any specific optimization measures taken.

**Questions:**

1. The third section of the paper devotes an entire section to discussing the motivation, clearly explaining the relevant content. However, it is debatable whether such an extensive section is necessary to elaborate on the motivation.
2. There appears to be a significant formatting issue at the bottom of page 3 and the top of page 4.

---

> ### Author Response · Authors · 2024-11-25
> **Response to reviewer (1/2)**
>
> We would like to thank the reviewer for valuable feedback and comments on our paper. We appreciate the opportunity to address your concerns and clarify any misunderstandings. Below, we provide detailed responses to each of your comments.
>
> > [W1]The paper's comparison of LoRA-X with other parameter-efficient fine-tuning methods is not comprehensive enough. I suggest the authors enhance the comparative analysis with existing methods, particularly the latest ones, to highlight the advantages of LoRA-X and potential areas for improvement.
>
> [A1] Below the table shows the performance of DoRA [1] and FouRA [2] on Trained and Transferred scenarios. ( We added new experiments to compare with DoRA/FouRA in Section 5.3 of the revised version.) From the results, we see that the projection idea works well on both these type of adapters. However, the DINO score of the transfer is relatively small compared to that of LoRA-X transfer. Moreover, the percentage change of transferred and trained adapters are higher suggesting that LoRA-X transfers better.
>
> |**Method**  |  **Adapter**  | **Rank** | **Dataset** | **HPSv2**                          | **LPIPS** |  **DINOv2** |
> |------------ |-------------|--------| ------------- |------------------------------------|---------------------| ----------- |
> | DoRA     | Trained        |   8    |    Paintings     | 0.3042                             | 0.4624              |     0.9138 |
> | | Transferred   | |     |                             0.2764 |      0.4526      |                     |
> |DoRA   |     Trained   |     8     |    Origami     |        0.2491      |    0.3408     | 0.9441 |
> | | Transferred  |        |       |    0.2224     |      0.3073 |  |
> | FouRA     | Trained    |    64     |  Paintings    |        0.3034  |  0.4686   |  0.9153 |
> | | Transferred  | |  |  0.2891  |  0.4446  |
>
>
> [1] Liu et al. Dora: Weight-decomposed low-rank adaptation. arXiv preprint arXiv:2402.09353, 2024.
>
> [2] Borse et al. FouRA: Fourier low rank adaptation. arXiv [cs.CV], 2024.
>
>
> > [W2] The paper introduces the cross-model adapter LoRA-X but does not emphasize which specific task the method focuses on. While the experimental section shows excellent performance in text-to-image generation tasks, its potential application in other domains is not fully explored. If the method is limited to a particular task, it would be beneficial to clarify this at the beginning of the paper to help readers understand better. Alternatively, if LoRA-X can be applied to multiple different application areas and tasks, a thorough discussion of its performance across various tasks would be valuable.
>
> [A2]  We appreciate the reviewer's suggestion. We have incorporated a LoRA-X application for fine-tuning TinyLlama (a large language model) and successfully transferred it to another version of TinyLlama for prompt generation tasks on "awesome chatgpt prompts" dataset. Please refer to the table comparing Bleu and Rouge metrics on the prompt generation task. We added the experiment in Appendix E.3 of the revised version. We include additional experiments on benchmark datasets in the camera ready version of the paper.
>
>
> |**Method** |  **Adapter** |  **Bleu ($\uparrow$)** |   **ROUGE-1 ($\uparrow$)** | **ROUGE-2 ($\uparrow$)**  | **ROUGE-L ($\uparrow$)** |  **ROUGE-LSum ($\uparrow$)** |
> |------------ | -------------| ----------------------- | -------------------------- | -------------------------- | -------------------------- | ----------------------------- |
>   |   LoRA-X    |   Trained |          0.8612        |            0.9349         |            0.9346          |           0.9349            |          0.9349 |
> | |   Transferred    |      0.8819           |         0.9874         |            0.9873          |           0.9874           |           0.9874 |

---

> > ### Author Response · Authors · 2024-11-25
> > **Response to reviewer (2/2)**
> >
> > > [W3]The paper's description of the implementation details of LoRA-X, including the specific implementation and optimization strategies of the algorithm, is not detailed enough. I recommend the authors provide more implementation details, including pseudocode or flowcharts of the algorithm, as well as any specific optimization measures taken.
> >
> > [A3] Thank you for pointing this out. Appendix B.1 and B.2 in the text describe our implementation details and hyperparameters. To these Sections, we will provide additional details such as the repositories we used as baseline. Apart from the hyperparameters mentioned, we used default settings from each repository. We provided ablation studies to detail the hyperparameters we tuned in (Below the table shows ablation on different hyper parameters on training LoRA-X using base model SD-v1.5 and BlueFire dataset. ) Appendix B.2. We also added a simplistic Pytorch psudocode in Appendix F of the revised paper.
> >
> > | **Steps** |  **Batch size** |  **HPSv2** |   **LPIPS** |
> > | ------------------ | ------------ | -------------------- | -------------------- |
> > | 5000    |     4      |        0.284     |           0.528  |
> > |    2000    |     4       |       0.260    |            0.518  |
> > |   2000   |      8      |        0.266       |         0.517  |
> > |  **5000**   |  **8**   |       **0.296**    |        **0.539** |
> >
> > > [Q1] The third section of the paper devotes an entire section to discussing the motivation, clearly explaining the relevant content. However, it is debatable whether such an extensive section is necessary to elaborate on the motivation.
> >
> > [A1] We revised the motivation section and added an illustration in Figure 2 of the revised version.
> >
> > > [Q2] There appears to be a significant formatting issue at the bottom of page 3 and the top of page 4.
> >
> > [A2] Thanks for informing us, it was due to the citation breaking between two pages and we resolved the issue in the revised version.

---

> ### Author Response · Authors · 2024-11-28
> **Response to reviewer**
>
> With reference to the following comment
>
> > [W1]The paper's comparison of LoRA-X with other parameter-efficient fine-tuning methods is not comprehensive enough. I suggest the authors enhance the comparative analysis with existing methods, particularly the latest ones, to highlight the advantages of LoRA-X and potential areas for improvement.
>
> We compared the performance of transferred LoRA-X using our training-free method based on equation (3) with X-Adapter [1], which uses plug-and-play modules trained on the target model. The table below shows the comparison: the "Transferred" row for LoRA-X indicates our training-free transfer from SSD-1B to SDXL, while for X-Adapter, it refers to the transfer method using X-adapter modules trained for SD-v1.5 to SDXL. The ``Trained'' row for both methods refers to trained LoRA-X adapter from scratch using BlueFire dataset.
>
> |**Method**  |  **Adapter**  |  **HPSv2**                          | **LPIPS** |  **DINOv2** |
> |------------ |-------------| ------------------------------------|---------------------| ----------- |
> | LoRA-X     | Trained        |    0.306                            | 0.422              |     0.953 |
> | | Transferred       |                             0.279 |      0.433      |                     |
> | X-Adapter  |     Trained   |              0.306      |    0.422     | 0.892 |
> | | Transferred        |    0.282     |      0.406 |  |
>
> Results show change in performance for HPSv2 & LPIPS from the trained baseline is in similar. However, our LoRA-X transfer produces higher DINO score mainly because it is transferred from a source in the similar family i.e SSD-1B. Also inference time for X-adapter is higher due to processing through base model, transferred model and the adapter.
>
> We have updated the results in the revised PDF and hope it answers your question and you would consider improving your evaluation.

---

> ### Author Response · Authors · 2024-11-30
> **Additional Results on NLP tasks**
>
> > [W2] The paper introduces the cross-model adapter LoRA-X but does not emphasize which specific task the method focuses on. While the experimental section shows excellent performance in text-to-image generation tasks, its potential application in other domains is not fully explored. If the method is limited to a particular task, it would be beneficial to clarify this at the beginning of the paper to help readers understand better. Alternatively, if LoRA-X can be applied to multiple different application areas and tasks, a thorough discussion of its performance across various tasks would be valuable.
>
> [A2]  We appreciate the reviewer's suggestion. We have incorporated a LoRA-X application for fine-tuning TinyLlama (a large language model) and successfully transferred it to another version of TinyLlama for more standard text generation tasks benchmarked in the original LoRA paper [1]. This include text-to-text generation on restaurant data (E2E NLG) [3] and on text summarization data (SamSum) [3]. For both of these tasks, we see small differences in Bleu and Rouge scores between the two models i.e. with LoRA-X transferred from source to target model and LoRA-X trained from scratch on the target model. The results confirm that our method can also be applied to other language tasks as well. All these results will be added into the camera ready submission.
>
> **Results on E2E-NLG Task:**
>
> |**Method** |  **Adapter** |  **Bleu ($\uparrow$)** |   **ROUGE-1 ($\uparrow$)** | **ROUGE-2 ($\uparrow$)**  | **ROUGE-L ($\uparrow$)** |  **ROUGE-LSum ($\uparrow$)** |
> |------------ | -------------| ----------------------- | -------------------------- | -------------------------- | -------------------------- | ----------------------------- |
>   |   LoRA-X    |   Trained |          0.6503        |            0.7689         |            0.6267          |           0.7533            |          0.7533 |
> | |   Transferred    |      0.6603           |         0.7661        |            0.6423          |           0.7624           |           0.7621 |
>
> **Results on SamSum Task:**
>
> |**Method** |  **Adapter** |   **ROUGE-1 ($\uparrow$)** | **ROUGE-2 ($\uparrow$)**  | **ROUGE-L ($\uparrow$)** |  **ROUGE-LSum ($\uparrow$)** |
> |------------ | -------------| -------------------------- | -------------------------- | -------------------------- | ----------------------------- |
>   |   LoRA-X    |   Trained |            0.3394         |            0.1394          |           0.2731           |          0.2731 |
> | |   Transferred    |         0.3568         |            0.1526          |           0.2884           |           0.2882 |
>
> References:
>
> [1] Hu, Edward J., et al. "LoRA: Low-Rank Adaptation of Large Language Models." International Conference on Learning Representations.2022
>
> [2] Novikova, Jekaterina, Ondřej Dušek, and Verena Rieser. "The E2E Dataset: New Challenges For End-to-End Generation." Proceedings of the 18th Annual SIGdial Meeting on Discourse and Dialogue. 2017.
>
> [3] Gliwa, Bogdan, et al. "SAMSum Corpus: A Human-annotated Dialogue Dataset for Abstractive Summarization." EMNLP-IJCNLP 2019 (2019): 70.
>
> **We hope these results address your comments regarding the evaluation on additional tasks. Our method has now been assessed on three NLP tasks. We hope these results will encourage you to reconsider the evaluation score as the discussion period is nearing its end.**

---

### Official Review · Reviewer_N9Bj · 2024-10-30

**Soundness:** 3
**Presentation:** 3
**Contribution:** 2
**Rating:** 6
**Confidence:** 3

**Summary:**

This study introduces LoRA-X to address the transferability problem of existing PEFT techniques across base models. It enables training-free cross-model adaptation by constraining the adapter within the source model’s subspace. LoRA-X demonstrates effective performance on text-to-image tasks, allowing seamless transfer without requiring original or synthetic data for retraining.

**Strengths:**

1. The introduction of LoRA-X as a training-free cross-model adapter addresses a significant limitation in PEFT. It allows adapters to be used across different base models without retraining or data access.

2. LoRA-X maintains a low parameter footprint, which is essential for computational efficiency.

**Weaknesses:**

1. The study only focuses on text-to-image tasks, which limits its applications to other domains such as NLP or time-series data.

2. LoRA-X has higher transfer costs when applied across significantly different architectures.

3. LoRA-X is only compared with the traditional LoRA. This could be strengthened by comparing LoRA-X's training-free transfer method with other recent PEFT techniques or knowledge distillation methods

**Questions:**

1. How well does LoRA-X perform across other types of tasks or domains, like NLP or time-series analysis? Would additional fine-tuning be needed to adapt it effectively?

2. How does LoRA-X perform relative to techniques like Trans-LoRA or SVDiff in terms of computational efficiency and performance? Could a direct comparison be provided?

3. Does the complexity of the transfer process increase significantly with larger models, and what optimizations could make it more scalable?

4. Is there a recommended threshold for subspace similarity that ensures effective transfer without sacrificing performance? How sensitive is LoRA-X to variations in this threshold?

---

> ### Author Response · Authors · 2024-11-25
> **Response to reviewer (1/2)**
>
> We would like to thank the reviewer for valuable feedback and comments on our paper. We appreciate the opportunity to address your concerns and clarify any misunderstandings. Below, we provide detailed responses to each of your comments.
>
> > [W1] The study only focuses on text-to-image tasks, which limits its applications to other domains such as NLP or time-series data.
>
> [A1] We appreciate the reviewer's suggestion. We have incorporated a LoRA-X application for fine-tuning TinyLlama (a large language model) and successfully transferred it to another version of TinyLlama for prompt generation tasks on "awesome chatgpt prompts" dataset. Please refer to the table comparing Bleu and Rouge metrics on the prompt generation task . We added the experiment in Appendix E.3 of the revised version. We include additional experiments on benchmark datasets in the camera ready version of the paper.
>
>
> |**Method** |  **Adapter** |  **Bleu ($\uparrow$)** |   **ROUGE-1 ($\uparrow$)** | **ROUGE-2 ($\uparrow$)**  | **ROUGE-L ($\uparrow$)** |  **ROUGE-LSum ($\uparrow$)** |
> |------------ | -------------| ----------------------- | -------------------------- | -------------------------- | -------------------------- | ----------------------------- |
>   |   LoRA-X    |   Trained |          0.8612        |            0.9349         |            0.9346          |           0.9349            |          0.9349 |
> | |   Transferred    |      0.8819           |         0.9874         |            0.9873          |           0.9874           |           0.9874 |
>
> > [W2] LoRA-X has higher transfer costs when applied across significantly different architectures.
>
> [A2] We fully agree with the reviewer. Indeed, LoRA-X cannot be transferred across significantly different architectures, as our goal is to achieve transfer without any training. We have introduced a transferability metric based on subspace similarity to indicate whether LoRA-X can be transferred from a source model to a target model without training. For example, in the case of SDXL and SD-v1.5, which have very different architectures, layers, hidden features, and number of heads, the ATC metric shows a very high value, indicating that training-free transfer is difficult between these two models.
>
> > [W3] LoRA-X is only compared with the traditional LoRA. This could be strengthened by comparing LoRA-X's training-free transfer method with other recent PEFT techniques or knowledge distillation methods.
>
> [A3] We appreciate the reviewer's comment. Our main objective is to design an adapter that can be transferred without training from a source model to a target model. To this end, we have added several experiments on the transferability of recent PEFT techniques, such as DoRA and FouRA, on the Origami and Painting.
> Below the table shows the performance of DoRA [1] and FouRA [2] on Trained and Transferred scenarios. ( We added new experiments to compare with DoRA/FouRA in Section 5.3 of the revised version.) From the results, we see that the projection idea works well on both these type of adapters. However, the DINO score of the transfer is relatively small compared to that of LoRA-X transfer. Moreover, the percentage change of transferred and trained adapters are higher suggesting that LoRA-X transfers better.
>
> |**Method**  |  **Adapter**  | **Rank** | **Dataset** | **HPSv2**                          | **LPIPS** |  **DINOv2** |
> |------------ |-------------|--------| ------------- |------------------------------------|---------------------| ----------- |
> | DoRA     | Trained        |   8    |    Paintings     | 0.3042                             | 0.4624              |     0.9138 |
> | | Transferred   | |     |                             0.2764 |      0.4526      |                     |
> |DoRA   |     Trained   |     8     |    Origami     |        0.2491      |    0.3408     | 0.9441 |
> | | Transferred  |        |       |    0.2224     |      0.3073 |  |
> | FouRA     | Trained    |    64     |  Paintings    |        0.3034  |  0.4686   |  0.9153 |
> | | Transferred  | |  |  0.2891  |  0.4446  |
>
>
> [1] Liu et al. Dora: Weight-decomposed low-rank adaptation. arXiv preprint arXiv:2402.09353, 2024.
>
> [2] Borse et al. FouRA: Fourier low rank adaptation. arXiv [cs.CV], 2024.

---

> > ### Author Response · Authors · 2024-11-25
> > **Response to reviewer (2/2)**
> >
> > > [Q1] How well does LoRA-X perform across other types of tasks or domains, like NLP or time-series analysis? Would additional fine-tuning be needed to adapt it effectively?
> >
> > [A1] Please refer to our reply on W1.
> >
> > > [Q2] How does LoRA-X perform relative to techniques like Trans-LoRA or SVDiff in terms of computational efficiency and performance? Could a direct comparison be provided?
> >
> > [A2] Trans-LoRA requires training using synthetic data to transfer a LoRA from the source to the target model. SVDiff modifies the singular values of the base model weights in all linear modules (Attention and Conv layers) and adjusts all the singular values. In contrast, LoRA-X is applied only to the Attention modules and uses a rank smaller than the base model's rank. The computational complexity for precalculating the SVD of the base model's weights for SVDiff is $O(MN \min(M,N))$, whereas for LoRA-X, it is $O(MNR)$, where R is the rank of the adapter.
> >
> > > [Q3] Does the complexity of the transfer process increase significantly with larger models, and what optimizations could make it more scalable?
> >
> > [A3] Our proposed transfer process requires the computation of SVD for each matrix which LoRA-X is applied to. However, as we fix the SVD rank R, the time complexity for each MxN matrix is O(MNR). Hence, the method is practically scalable. Additionally, it is much lesser complex compared to training a new LoRA on the Target model, as the Source model LoRA can be transferred without training to all models of the same family. We will mention this in the revised text.
> >
> > > [Q4] Is there a recommended threshold for subspace similarity that ensures effective transfer without sacrificing performance? How sensitive is LoRA-X to variations in this threshold?
> >
> > [A4] In our analysis in text-to-image generation task, we observed that subspace similarity above 0.8 is sufficient but it could be model and task dependent.

---

> > > ### Comment · Reviewer_N9Bj · 2024-11-26
> > > **No further comments**
> > >
> > > Dear Author(s),
> > >
> > > Thank you for addressing my concerns. I have no further questions.

---

> ### Author Response · Authors · 2024-11-30
> **Additional Results on NLP tasks**
>
> > [W1] The study only focuses on text-to-image tasks, which limits its applications to other domains such as NLP or time-series data.
>
> [A1]  We appreciate the reviewer's suggestion. We have incorporated a LoRA-X application for fine-tuning TinyLlama (a large language model) and successfully transferred it to another version of TinyLlama for more standard text generation tasks benchmarked in the original LoRA paper [1]. This include text-to-text generation on restaurant data (E2E NLG) [3] and on text summarization data (SamSum) [3]. For both of these tasks, we see small differences in Bleu and Rouge scores between the two models i.e. with LoRA-X transferred from source to target model and LoRA-X trained from scratch on the target model. The results confirm that our method can also be applied to other language tasks as well. All these results will be added into the camera ready submission.
>
> **Results on E2E-NLG Task:**
>
> |**Method** |  **Adapter** |  **Bleu ($\uparrow$)** |   **ROUGE-1 ($\uparrow$)** | **ROUGE-2 ($\uparrow$)**  | **ROUGE-L ($\uparrow$)** |  **ROUGE-LSum ($\uparrow$)** |
> |------------ | -------------| ----------------------- | -------------------------- | -------------------------- | -------------------------- | ----------------------------- |
>   |   LoRA-X    |   Trained |          0.6503        |            0.7689         |            0.6267          |           0.7533            |          0.7533 |
> | |   Transferred    |      0.6603           |         0.7661        |            0.6423          |           0.7624           |           0.7621 |
>
> **Results on SamSum Task:**
>
> |**Method** |  **Adapter** |   **ROUGE-1 ($\uparrow$)** | **ROUGE-2 ($\uparrow$)**  | **ROUGE-L ($\uparrow$)** |  **ROUGE-LSum ($\uparrow$)** |
> |------------ | -------------| -------------------------- | -------------------------- | -------------------------- | ----------------------------- |
>   |   LoRA-X    |   Trained |            0.3394         |            0.1394          |           0.2731           |          0.2731 |
> | |   Transferred    |         0.3568         |            0.1526          |           0.2884           |           0.2882 |
>
> References:
>
> [1] Hu, Edward J., et al. "LoRA: Low-Rank Adaptation of Large Language Models." International Conference on Learning Representations.2022
>
> [2] Novikova, Jekaterina, Ondřej Dušek, and Verena Rieser. "The E2E Dataset: New Challenges For End-to-End Generation." Proceedings of the 18th Annual SIGdial Meeting on Discourse and Dialogue. 2017.
>
> [3] Gliwa, Bogdan, et al. "SAMSum Corpus: A Human-annotated Dialogue Dataset for Abstractive Summarization." EMNLP-IJCNLP 2019 (2019): 70.

---

### Official Review · Reviewer_ZEzP · 2024-11-03

**Soundness:** 3
**Presentation:** 2
**Contribution:** 2
**Rating:** 6
**Confidence:** 4

**Summary:**

This paper addresses the binding issue between LoRA models and their base models in data-scarce scenarios. It introduces Cross-Model Low-Rank Adaptation (LoRA-X), an adapter that operates within the subspace of pre-trained diffusion models to facilitate style transfer from the source base model to the target base model. Qualitative and quantitative experiments demonstrate its effectiveness.

**Strengths:**

1.  This is a meaningful work, addressing a common industry limitation: the need to retrain LoRA when changing the pre-trained model.

2. The paper introduces a subspace perspective for Stable Diffusion models, which may inspire future research.

**Weaknesses:**

1. The paper lacks comparisons with recent works, such as [1].

2. It is better to conduct some visualization to illustrate the motivation. This may better show the effects of the same LoRA model combined with different base models to highlight this pressing challenge.

3. The experiments appear overly simplistic:

(1)  There is no baseline; for models based on SD 1.5, SD 1.5 should serve as the baseline, and the same applies to SDXL.

(2)  In Section 5.2, there is a lack of quantitative analysis for the target model without LoRA-X and for [1]. Additionally, the source row seems redundant and could be placed in later experiments, as the main experiment only needs to compare different methods.

(3)  In Section 5.3, at least two to three datasets are needed to compare the performance drop of the LoRA-X style transfer method against LoRA. Furthermore, the ranks of LoRA-X and LoRA are not the same; comparisons under the same rank are missing.

(4)  There is no experiment on the impact of rank on LoRA-X performance.

(5)  In all tables, "Results in Green and Red show less than and more than 10% difference" should be replaced with ± percentage for clarity.

4. The writing details need further refinement, as noted in the Questions, to avoid reader confusion.

 [1] Ran, Lingmin, et al. "X-adapter: Adding universal compatibility of plugins for upgraded diffusion model." Proceedings of the IEEE/CVF Conference on Computer Vision and Pattern Recognition. 2024.

**Questions:**

Q1: The term "base model" is unclear in the paper. If the source model is SD 1.5, does the target model refer to the pre-trained model fine-tuned based on SD 1.5 or SD XL? Although the experiments indicate that the latter is challenging and lack corresponding quantitative analysis, this should be clarified in the introduction, specifying that it refers to the former.

Q2: In Figure 2, the phrase "without access to the original data" in the introduction suggests that the target model (b) does not require training, but the source model (a) still needs data. Similar to Q1, the semantics should be clarified in the text.

Q3: In Section 4.2.2, the phrase "linear transformation can be evaluated as P" raises questions about the relationship between P and the subsequent formula U_s . Why is P mentioned?

Q4: In Section 5.3, the statement "We repeated the experiment for different LoRA ranks to show how LoRA’s transferability drops as rank is reduced, though its total size remains much higher than LoRA-X" needs clarification on which specific metric in Table 2 illustrates this transferability.

Q5: In Section 5.4.2, the \Delta \Sigma_s row is not represented in Table 4.

---

> ### Author Response · Authors · 2024-11-25
> **Response to reviewer (1/3)**
>
> We would like to thank the reviewer for valuable feedback and comments on our paper. We appreciate the opportunity to address your concerns and clarify any misunderstandings. Below, we provide detailed responses to each of your comments.
> > [W1] The paper lacks comparisons with recent works, such as [1].
>
> [A1] We appreciate the reviewer's suggestion to consider the paper [1] on X-Adapter. This work introduces a universal mapper for transferring adapters between diffusion models, and we will certainly cite it (we added in Section 2 of the revised version). However, we believe a direct comparison with X-Adapter is not appropriate. Our approach leverages LoRA, while X-Adapter employs the IP-adapter method, which is not a PEFT method and incurs additional inference costs by doubling the number of cross-attentions in the network. Moreover, X-Adapter requires training for each target model using a shared dataset subset, whereas our method offers a training-free transfer of LoRA-X across text-to-image diffusion models.
>
> > [W2] It is better to conduct some visualization to illustrate the motivation. This may better show the effects of the same LoRA model combined with different base models to highlight this pressing challenge.
>
> [A2] Table 2 of the main text discusses the effect of LoRA transfer v/s LoRA-X transfer. As rightly pointed out, we will illustrate this effect by showing qualitative examples of LoRA transfer v/s our proposed approach, to better explain our motivation. We agree that this suggestion will help us improve the work. We added the illustration in Figure 2.b of the revised version.
>
> > [W3] The experiments appear overly simplistic:
>
> > [W3.1] There is no baseline; for models based on SD 1.5, SD 1.5 should serve as the baseline, and the same applies to SDXL.
>
> [A3.1] We renamed the 'Source' and 'Target' labels in the Experiment Section to 'Trained' and 'Transferred,' respectively, to avoid confusion. As discussed in Tables 1 and 2 of the revised version, we compare our method against two baselines. In Table 1, we report the performance of LoRA-X on the "Transferred" (our proposed approach) versus the "Trained".
> * "Trained": The LoRA-X adapter is trained on the base model (SD Eff-v1.0) from scratch using a training dataset. We then generate samples using the combined base model (SD Eff-v1.0) and the trained adapter.
> * "Transferred": The LoRA-X adapter is transferred from another source model's adapter (SD-v1.5) without any additional training. We then generate samples using the combined base model (SD Eff-v1.0) and the transferred adapter.
>
> The "Trained" serves as the baseline (or upper bound), and we expect the evaluated metric on the "Transferred" to be close to that of the "Trained." Additionally, in Table 2, we use LoRA as a baseline, comparing the transfer of LoRA trained on a source to a target, versus LoRA-X on the same source-to-target combination. Furthermore, we added Table 3 in the revised version to compare LoRA-X Trained and Transferred with the ones using DoRA and FouRA.
>
>
> > [W3.2]In Section 5.2, there is a lack of quantitative analysis for the target model without LoRA-X and for [1]. Additionally, the source row seems redundant and could be placed in later experiments, as the main experiment only needs to compare different methods.
>
> [A3.2] We would like to thank the reviewer for bringing up this paper. For the question on [1], please refer to our reply at [A1]. For the question on "Source" ("Trained" in the revised version), we use it as a baseline (or upper bound) to achieve a similar result with our training-free approach. It is necessary as it acts as a baseline.

---

> > ### Author Response · Authors · 2024-11-25
> > **Response to reviewer (2/3)**
> >
> > > [W3.3] In Section 5.3, at least two to three datasets are needed to compare the performance drop of the LoRA-X style transfer method against LoRA. Furthermore, the ranks of LoRA-X and LoRA are not the same; comparisons under the same rank are missing.
> >
> > [A3.3] We appreciate the reviewer's suggestion. We added the ablation experiment for the Origami dataset as well in Appendix E.1 of the revised paper. We will add the similar ablation on Painting in the camera ready version of the paper.
> > Regarding the comparison between LoRA and LoRA-X at the same rank, it’s important to note that both are PEFT methods designed to reduce the number of parameters needed for fine-tuning downstream tasks. However, we believe this comparison is not entirely appropriate, as the number of parameters fine-tuned in LoRA is significantly higher than in LoRA-X. As shown in the table below, LoRA with a rank of 32 has a much larger parameter size compared to LoRA-X with a rank of 320. This is because LoRA-X modifies only a subset of the singular values of the base model’s weights.
> >
> > |**Dataset**  |  **Method** |  **Adapter**  | **Rank**  | **HPSv2** |  **LPIPS**  | **DINOv2** |  **Total size (MB)**|
> >  | ------------- | ------------ | ------------- | ---------- | ------------------------ | ---------------------------------- | ------------------------- |--------------------- |
> >  | Origami     |     LoRA-X   |    Trained   |    320      |        0.265        |                 0.521         |               **0.819**       |          0.16 |
> >  |                   |        |  Transferred     |          |       0.330          |               0.484       |                      |             |
> > |                 |   LoRA    |    Trained   |     32     |         0.253       |                  0.414      |                    0.812         |          34.07 |
> >   |              |            |  Transferred      |           |     0.226         |                0.482    |                 |                  |
> >  |            |             |      Trained  |      16      |        0.261         |                0.460      |                    0.781        |           17.08 |
> >   |             |            |   Transferred       |         |      0.229         |                0.475         |                 |                |
> >   |            |              |    Trained   |     1       |        0.255       |                  0.480        |                  0.798        |           1.15 |
> >   |              |           |   Transferred      |           |     0.230        |                 0.492      |                    |                  |
> >
> > > [W3.4] There is no experiment on the impact of rank on LoRA-X performance.
> >
> > [A3.4] We appreciate the reviewer's suggestion. We added an experiment, in Section 5.4.2 of the revised version, to evaluate the performance of LoRA-X at different ranks, which refers to the number of singular values modified. As shown in the table below, the performance of LoRA-X (Trained rows, trained on SD Eff-v1.0) decreases as the rank drops. However, the performance of the transferred LoRA-X (Transferred rows, transferred from SD-v1.5) remains close to that of the Trained version.
> >
> > |**Method** |  **Adapter**  | **Rank**  | **HPSv2**    |  **LPIPS** |  **DINOv2** |  **Total size (MB)** |
> >  | ------------| ------------- |--------------| ------------------------ |----------------------------------| ------------------------- |---------------------|
> >   |   LoRA-X    |   Trained     |  320     | 0.2958       |                    0.5340      |                  0.8513    |               0.16 |
> >   |          |    Transferred    |          | 0.3073       |                  0.5376      |              |                    |
> >    |            |   Trained    |   160     | 0.2850       |                 0.5310       |                 0.8352    |                0.1 |
> >    |          |   Transferred   |            | 0.2849       |                 0.5263     |              |                  |
> >    |           |    Trained    |    80      | 0.2782       |                  0.5294    |                    0.8300     |              0.05 |
> >    |           |  Transferred       |         |       0.2723 |                 0.5224    |                 |
> >
> > > [W3.5] In all tables, "Results in Green and Red show less than and more than 10\% difference" should be replaced with ± percentage for clarity.
> >
> > [A3.5] We appreciate the reviewer suggestion. We updated all the tables in the revised version.
> >
> > > [W4] The writing details need further refinement, as noted in the Questions, to avoid reader confusion.
> >
> > [A4] We appreciate the reviewer's suggestion. We have made the necessary refinements in the revised version.

---

> > > ### Author Response · Authors · 2024-11-25
> > > **Response to reviewer (3/3)**
> > >
> > > > [Q1] The term "base model" is unclear in the paper. If the source model is SD 1.5, does the target model refer to the pre-trained model fine-tuned based on SD 1.5 or SD XL? Although the experiments indicate that the latter is challenging and lack corresponding quantitative analysis, this should be clarified in the introduction, specifying that it refers to the former.
> > >
> > > [A1] In our paper, the term "base model" refers to a pre-trained model. The "source model" is a base model with an adapter that has been trained from scratch using a training dataset. The "target model" is another base model with an adapter transferred from the source model without any training. We added the explanation of definition of Target and Source at the end of Introduction Section in the revised version.
> > >
> > > > [Q2] In Figure 2, the phrase "without access to the original data" in the introduction suggests that the target model (b) does not require training, but the source model (a) still needs data. Similar to Q1, the semantics should be clarified in the text.
> > >
> > > [A2] We thank the review for the comment and to clarify the naming we replaced Source and Target in Tables by Trained and Transferred, respectively. We updated all the tables in the revised version, accordingly. Please refer to our reply to W3.1 for definition of the terms.  We consider LoRA-X to be transferable if the metrics (such as HPSv2 and LPIPS) evaluated on the generated samples for these two scenarios are similar, and if the DinoV2 features extracted from the samples in both scenarios are highly correlated.
> > >
> > > > [Q3] In Section 4.2.2, the phrase "linear transformation can be evaluated as P" raises questions about the relationship between P and the subsequent formula $U_s$ . Why is $P$ mentioned?
> > >
> > > [A3] The projection $P$ is necessary when $U_s$ and $U_t$ have different number of rows $m \neq m'$. In this case, we cannot directly use equation (3). Instead, we need to project $U_s$ onto the common (row) subspace of $U_t$. The projection $P$ performs this task. After projecting $U_s$, we obtain $\tilde{U}_s = PU_s$, which has the same number of rows as $U_t$. We can then apply equation (3). We added clarification in the revised version Section 4.2.2.
> > >
> > > > [Q4] In Section 5.3, the statement "We repeated the experiment for different LoRA ranks to show how LoRA’s transferability drops as rank is reduced, though its total size remains much higher than LoRA-X" needs clarification on which specific metric in Table 2 illustrates this transferability.
> > >
> > > [A4] As mentioned in our response to Q2, we consider two scenarios: the "Trained case," where the adapter is trained from scratch using a training dataset on a specific base model, and the "Transferred case," where the adapter is transferred from a source adapter of a different base model and applied on the same base model as the Trained case. We expect the metrics for these two cases to be similar, indicating a successful transfer.
> > >
> > > > [Q5] In Section 5.4.2, the $\Delta \Sigma_s$ row is not represented in Table 4.
> > >
> > > [A5] The $\Delta \Sigma_s$ row should be as "source row." We corrected this in the revised version.

---

> > > > ### Comment · Reviewer_ZEzP · 2024-11-27
> > > >
> > > > Thanks for the authors' response and explanation. While many of my concerns have been addressed and I also acknowledge its difference from other methods,  I still believe that my main concern, i.e. a comparison with X-Adapter, is necessary, as both methods target very similar application scenarios. Although X-Adapter may incur higher resource consumption and inference costs, quantitatively comparing key metrics such as performance and resource efficiency between the two methods would provide  more comprehensive information for the area. Such a comparison would allow readers to make more informed decisions based on their specific needs and fairly highlight the strengths and weaknesses of each approach.
> > > >
> > > > In light of these reasons, I am inclined to keep my previous rating for the moment. I will nonetheless discuss my evaluation with the other fellow reviewers  to reach my final recommendation.

---

> > > > > ### Author Response · Authors · 2024-11-27
> > > > > **Response to Reviewer ZEzP**
> > > > >
> > > > > We are pleased to know that many of your concerns have been addressed. We thank you for your prompt response in informing that the concern with X-adapter still remains. We hope to address this concern
> > > > > within the discussion period by performing additional comparison studies with X-adapter. Having said that we have the following observations:
> > > > >
> > > > > (a) X-adapter codebase currently is inference only and can only cater to transferring LoRAs from SD1.5 to SDXL. Currently, training script is not available to train the X-adapter across other models
> > > > > i.e. SSD-1B to SDXL and vice versa.
> > > > >
> > > > > (b) While X-adapter can consider training-based transfer between different family of diffusion models such as from SD1.5 to SDXL, our method can consider training-free transfer within same family ie. SDXL to SSD-1B or SD1.5 to Effv1.
> > > > >
> > > > > So, for fair comparison with X-adapter its difficult to keep both the source (SD1.5) and target (SDXL) the same. However, we can keep the target same i.e. SDXL but vary the source (i.e SSD/RVXL for our method) to keep
> > > > > it within the scope of LoRA-X transfer. We wanted to confirm whether such a comparison study would be beneficial and you would recommend for a higher evaluation.

---

> > > > > > ### Comment · Reviewer_ZEzP · 2024-11-27
> > > > > >
> > > > > > Thanks for your quick response. I understand the difficulty in comparing X-adapter directly.  Your proposed  plan sounds reasonable to me. I would also suggest the authors add in the revision more discussions with X-adapter both analytically and empirically (if possible) to enhance the impact of the paper.

---

> ### Author Response · Authors · 2024-11-28
> **Response to Reviewer ZEzP**
>
> Thanks for accepting the suggestion. We compared the performance of transferred LoRA-X using our training-free method based on equation (3) with X-Adapter [1], which uses plug-and-play modules trained on the target model. The table below shows the comparison: the "Transferred" row for LoRA-X indicates our training-free transfer from SSD-1B to SDXL, while for X-Adapter, it refers to the transfer method using X-adapter modules trained for SD-v1.5 to SDXL. The ``Trained'' row for both methods refers to trained LoRA-X adapter from scratch using BlueFire dataset.
>
> |**Method**  |  **Adapter**  |  **HPSv2**                          | **LPIPS** |  **DINOv2** |
> |------------ |-------------| ------------------------------------|---------------------| ----------- |
> | LoRA-X     | Trained        |    0.306                            | 0.422              |     0.953 |
> | | Transferred       |                             0.279 |      0.433      |                     |
> | X-Adapter  |     Trained   |              0.306      |    0.422     | 0.892 |
> | | Transferred        |    0.282     |      0.406 |  |
>
> Results show change in performance for HPSv2 & LPIPS from the trained baseline is in similar. However, our LoRA-X transfer produces higher DINO score mainly because it is transferred from a source in the similar family i.e SSD-1B. Also inference time for X-adapter is higher due to processing through base model, transferred model and the adapter.
>
> We have updated the results in the revised PDF and hope it answers your question regarding analytical and empirical comparison.

---

> > ### Comment · Reviewer_ZEzP · 2024-11-28
> >
> > Thanks for the additional information! I think the authors have well addressed my concerns. Therefore, I am happy to raise my rating accordingly.

---

### Official Review · Reviewer_GaaU · 2024-11-03

**Soundness:** 3
**Presentation:** 3
**Contribution:** 3
**Rating:** 8
**Confidence:** 3

**Summary:**

The paper addresses the challenges associated with the fine-tuning of large foundation models, particularly in the context of Low-Rank Adaptation (LoRA). It highlights the complications that arise when base models are deprecated, necessitating the retraining of LoRA modules without access to original training data due to privacy or licensing constraints. To mitigate these issues, the authors propose a novel adapter that facilitates the transfer of LoRA parameters between source and target models without requiring original or synthetic training data. The method operates within the subspace of the source model and focuses on layers of the target model that demonstrate sufficient subspace similarity. The effectiveness of the proposed method is validated through extensive experiments in text-to-image generation tasks.

**Strengths:**

1. The introduction of LoRA-X presents a significant advancement in parameter-efficient fine-tuning by enabling the transfer of LoRA parameters without the need for retraining, addressing a critical gap in the existing methodologies. The solution is highly relevant in real-world applications where access to original training data is often restricted, making it a valuable contribution to the field.
2. The paper includes extensive experiments demonstrating the effectiveness of LoRA-X across multiple models and tasks, providing strong empirical support for the proposed method.

**Weaknesses:**

1. The paper should discuss the scalability of LoRA-X for larger models or more complex tasks beyond text-to-image generation.
2. The reliance on subspace similarity may restrict the applicability of LoRA-X to models that are closely related, potentially limiting its use in more diverse model architectures.

**Questions:**

See the weaknesses for details.

---

> ### Author Response · Authors · 2024-11-25
>
> We would like to thank the reviewer for valuable feedback and comments on our paper. We appreciate the opportunity to address your concerns and clarify any misunderstandings. Below, we provide detailed responses to each of your comments.
> >  The paper should discuss the scalability of LoRA-X for larger models or more complex tasks beyond text-to-image generation.
>
> [A1] We appreciate the reviewer's suggestion. We have incorporated a LoRA-X application for fine-tuning TinyLlama (a large language model) and successfully transferred it to another version of TinyLlama for prompt generation tasks on "awesome chatgpt prompts" dataset. Please refer to the table below comparing Bleu and Rouge metrics on the prompt generation task. We added the experiment in Appendix E.3 of the revised version. We include additional experiments on benchmark datasets in the camera ready version of the paper.
>
>
> |**Method** |  **Adapter** |  **Bleu ($\uparrow$)** |   **ROUGE-1 ($\uparrow$)** | **ROUGE-2 ($\uparrow$)**  | **ROUGE-L ($\uparrow$)** |  **ROUGE-LSum ($\uparrow$)** |
> |------------ | -------------| ----------------------- | -------------------------- | -------------------------- | -------------------------- | ----------------------------- |
>   |   LoRA-X    |   Trained |          0.8612        |            0.9349         |            0.9346          |           0.9349            |          0.9349 |
> | |   Transferred    |      0.8819           |         0.9874         |            0.9873          |           0.9874           |           0.9874 |
>
>
>
> > The reliance on subspace similarity may restrict the applicability of LoRA-X to models that are closely related, potentially limiting its use in more diverse model architectures.
>
> [A2] We fully agree that subspace similarity as a precondition limits the transfer of the adapter without additional training. However, there are numerous instances, particularly in text-to-image diffusion models, where the source and target models meet this precondition. In such cases, the alignment of subspaces allows for effective transfer, enabling the adapter to function well in the target model without the need for retraining.

---

> ### Author Response · Authors · 2024-11-30
> **Additional Results on NLP tasks**
>
> > [W1] The paper should discuss the scalability of LoRA-X for larger models or more complex tasks beyond text-to-image generation
>
> [A1]  We appreciate the reviewer's suggestion. We have incorporated a LoRA-X application for fine-tuning TinyLlama (a large language model) and successfully transferred it to another version of TinyLlama for more standard text generation tasks benchmarked in the original LoRA paper [1]. This include text-to-text generation on restaurant data (E2E NLG) [3] and on text summarization data (SamSum) [3]. For both of these tasks, we see small differences in Bleu and Rouge scores between the two models i.e. with LoRA-X transferred from source to target model and LoRA-X trained from scratch on the target model. The results confirm that our method can also be applied to other language tasks as well. All these results will be added into the camera ready submission.
>
> **Results on E2E-NLG Task:**
>
> |**Method** |  **Adapter** |  **Bleu ($\uparrow$)** |   **ROUGE-1 ($\uparrow$)** | **ROUGE-2 ($\uparrow$)**  | **ROUGE-L ($\uparrow$)** |  **ROUGE-LSum ($\uparrow$)** |
> |------------ | -------------| ----------------------- | -------------------------- | -------------------------- | -------------------------- | ----------------------------- |
>   |   LoRA-X    |   Trained |          0.6503        |            0.7689         |            0.6267          |           0.7533            |          0.7533 |
> | |   Transferred    |      0.6603           |         0.7661        |            0.6423          |           0.7624           |           0.7621 |
>
> **Results on SamSum Task:**
>
> |**Method** |  **Adapter** |   **ROUGE-1 ($\uparrow$)** | **ROUGE-2 ($\uparrow$)**  | **ROUGE-L ($\uparrow$)** |  **ROUGE-LSum ($\uparrow$)** |
> |------------ | -------------| -------------------------- | -------------------------- | -------------------------- | ----------------------------- |
>   |   LoRA-X    |   Trained |            0.3394         |            0.1394          |           0.2731           |          0.2731 |
> | |   Transferred    |         0.3568         |            0.1526          |           0.2884           |           0.2882 |
>
> References:
>
> [1] Hu, Edward J., et al. "LoRA: Low-Rank Adaptation of Large Language Models." International Conference on Learning Representations.2022
>
> [2] Novikova, Jekaterina, Ondřej Dušek, and Verena Rieser. "The E2E Dataset: New Challenges For End-to-End Generation." Proceedings of the 18th Annual SIGdial Meeting on Discourse and Dialogue. 2017.
>
> [3] Gliwa, Bogdan, et al. "SAMSum Corpus: A Human-annotated Dialogue Dataset for Abstractive Summarization." EMNLP-IJCNLP 2019 (2019): 70.

---

### Author Response · Authors · 2024-11-25
**Summary of reviews**

We would like to thank all the reviewers for reviewing our paper and providing valuable and constructive feedback.
We are grateful that the reviewers have highlighted our work as:

* LoRA-X enables training-free, parameter-efficient cross model adaptation (reviewers: GaaU, ZEzP, N9Bj, 7APq)
* Theoretical analysis provides a solid foundation and comparison with other PEFT methods (reviewer: 7APq)
* Addresses real-world issues like data privacy by adapting models without original training data. (reviewers: GaaU, 7APq)
* Low parameter footprint ensures computational efficiency. (reviewer: N9Bj)
* Potential for future research, especially in Stable Diffusion models and style transfer. (reviewer: ZEzP)


To summarize the major responses we have made in rebuttal:
* **Performance Analysis**: We added LoRA-X performance analysis on the text generation task using TinyLlama.
* **Comparative Study**: We included a comparison of LoRA-X with LoRA (different ranks) on the Origami dataset.
* **Ablation Study**: We conducted an ablation study on different ranks for LoRA-X.
* **Comparison with Recent Methods**: We compared LoRA-X with the most recent PEFT methods, such as DoRA [1] and FouRA [2], including their transferred versions.
* **Hyperparameter Ablation**: We added an ablation study on different hyperparameters, including batch size and steps, modified from the repository.

Revised part is showed in blue in the revised version.
Again, we genuinely appreciate the input from reviewers and we thank all reviewers for their time and effort.

[1] Liu et al. Dora: Weight-decomposed low-rank adaptation. arXiv preprint arXiv:2402.09353, 2024.

[2] Borse et al. FouRA: Fourier low rank adaptation. arXiv [cs.CV], 2024.

---

### Meta-Review · Area_Chair_XZNx · 2024-12-20

**Metareview:**

The paper presents a significant advancement in the fine-tuning of large foundation models by addressing the challenges associated with Low-Rank Adaptation (LoRA), particularly when base models are deprecated and original training data is inaccessible due to privacy or licensing constraints. The authors propose LoRA-X, a novel adapter that enables the transfer of LoRA parameters between source and target models without requiring retraining or access to original or synthetic data. This method operates within the subspace of the source model and targets layers with sufficient subspace similarity in the target model. Extensive experiments, particularly in text-to-image generation tasks, validate the effectiveness of LoRA-X, demonstrating its potential to overcome a critical limitation in current methodologies. The introduction of a subspace perspective for Stable Diffusion models not only addresses a common industry challenge but also opens avenues for future research, making this work a valuable contribution to the field.

**Additional Comments On Reviewer Discussion:**

After the rebuttal, most of the reviewers expressed positive opinions (except for one reviewer who did not provide further response), and one reviewer increased their score accordingly.

---

### Decision · Program_Chairs · 2025-01-22

Accept (Poster)